



**Microbial activity, methane production, and carbon storage in Early Holocene North Sea peats**

Tanya J.R. Lippmann[1†], Michiel H. in 't Zandt[2,3†], Nathalie N.L. Van der Putten[1], Freek S. Busschers[4], Marc P. Hijma[5], Pieter van der Velden[3], Tim de Groot[6], Zicarlo van Aalderen[1], Ove H. Meisel[1,3], Caroline P. Slomp[3,7], Helge Niemann[6,7,8], Mike S.M. Jetten[2,3,9], Han A.J. Dolman[1,3], Cornelia U. Welte[2,9]

[1] Department of Earth Sciences, Vrije Universiteit Amsterdam, De Boelelaan 1085, 1081 HV Amsterdam, the Netherlands

[2] Department of Microbiology, Institute for Water and Wetland Research, Radboud University, Heyendaalseweg 135, 6525 AJ Nijmegen, the Netherlands

[3] Netherlands Earth System Science Center, Utrecht University, Heidelberglaan 2, 3584 CS Utrecht, the Netherlands

[4] TNO - Geological Survey of the Netherlands, Princetonlaan 6, 3508 TA, Utrecht, The Netherlands

[5] Department of Applied Geology and Geophysics, Deltares Research Institute, Daltonlaan 600, 3584 BK Utrecht, the Netherlands

[6] Department of Microbiology & Biogeochemistry, Royal Netherlands Institute for Sea Research, Landsdiep 4, 1797 SZ 't Horntje, the Netherlands

[7] Department of Earth Sciences, Faculty of Geosciences, Utrecht University, Princetonlaan 8a 3584 CB Utrecht, the Netherlands

[8] Centre for Arctic Gas Hydrate (CAGE), Environment and Climate, Department of Geosciences, UiT The Arctic
University of Norway in Tromsø, Tromsø, Norway

[9] Soehngen Institute of Anaerobic Microbiology, Radboud University, Heyendaalseweg 135, 6525 AJ Nijmegen, the Netherlands

*Correspondence to*: Tanya J.R. Lippmann (t.j.r.lippmann@vu.nl)

†These authors contributed equally to this work.





**Abstract.**

Northern latitude peatlands act as important carbon sources and sinks but little is known about the greenhouse

gas (GHG) budget of peatlands submerged beneath the North Sea during the last glacial-interglacial transition.

We found that whilst peat formation was diachronous, commencing between 13,680 and 8,360 calibrated years before the present, stratigraphic layering and local vegetation succession were consistent across a large study area. The $CH_4$ concentrations of the sediment pore waters were low at most sites, with the exception of two

locations, and the stored carbon was large.

Incubation experiments in the laboratory revealed molecular signatures of methanogenic archaea, with strong increases in rates of activity upon methylated substrate amendment. Remarkably, methanotrophic activity and the respective diagnostic molecular signatures could be not be detected. Heterotrophic Bathyarchaeia dominated the

archaeal communities and bacterial populations were dominated by candidate phylum JS1 bacteria.

Although $CH_4$ accumulation is low at most sites, the presence of *in situ* methanogenic micro-organisms, the absence of methanotrophy, and large widespread stores of carbon hold the potential for GHG production if catalysed by a change in environmental conditions. Despite being earmarked as a critical source of $CH_4$ seepage,

seepage from these basal-peat deposits is restricted, as evidenced by low *in situ* $CH_4$ concentrations.



## 1 Introduction

The expansion and submersion of Northern latitude peatlands play a key role in global methane ($CH_4$) and carbon (C) cycles (e.g., (Charman et al., 2013; Morris et al., 2018)). Globally, peatlands serve as long-term carbon sinks
(Clymo et al., 1998; Gorham, 1991) that store more carbon than the world's forests combined, despite covering only 3% of the world's surface land area (Xu et al., 2018). At the time of the Last Glacial Maximum, peatlands worldwide stored 600,000 Tg C (Yu et al., 2010). This estimate is based on assumptions of peat layer thickness and depth at the ocean basin scale, but few *in situ* observations of peat deposit properties are available to verify these assumptions.

Methane is globally the second most prevalent greenhouse gas, with emissions to the atmosphere amounting to 550-594 Tg $CH_4$ each year (Saunois et al., 2020). Continental shelves and deltas are important sinks within the global carbon cycle (Oppo et al., 2020; Saunois et al., 2020) and are responsible for 80-85% of oceanic carbon sequestration (Muller-Karger et al., 2005). Shelf regions contribute ~75% of global ocean $CH_4$ flux to the atmosphere, with estimates of seep from oceanic shelves into bottom waters ranging between 6–12 Tg $CH_4$ $yr^{-1}$
(Weber et al., 2019), and 16-48 Tg $CH_4$ $yr^{-1}$ (Judd et al., 2002). Reducing the uncertainties in these estimates requires further work at both regional and global scales (Oppo et al., 2020; Saunois et al., 2020). The high $CH_4$ concentrations in surface waters of continental shelves are due to $CH_4$ inputs from estuaries and sea floor sediments, where methanogenesis is fuelled by high organic matter sedimentation (Carr et al., 2018; Zhuang et al., 2018). Methane entering the water column from the sea floor is released either as bubbles or by pore water
diffusion and can be of either biogenic or thermogenic origin. Variations in atmospheric $CH_4$ are due, in part, to the changing extent of peatlands over glacial interglacial periods (Frolking and Roulet, 2007).

Triggered by post-glacial sea-level rise and, consequently, rising ground water, peatlands in the area between the Netherlands, the United Kingdom and Denmark, now the North Sea, developed by the process of paludification in the Late Pleistocene and Early Holocene (Fig. 1A). Large parts of the tectonically subsiding North Sea basin
become dry during glacials and flooded during interglacials (Hijma et al., 2012). During the last ice age, ice sheets reaching as far south as the Doggerbank area were subjected to strong glacio-isostatic adjustment (GIA) (Vink et al., 2007). During the Early Holocene, GIA resulted in subsidence and, combined with rapid melting of global ice sheets, high rates of sea-level rise, up to 1-2 cm/year (Hijma and Cohen, 2019), giving rise to peatland development and later peatland submersion. As a result, former freshwater peatland ecosystems, preserved as a stiff peat layer
(basal-peat), became submerged marine ecosystems at the base of the Early Holocene sequence.

The establishment of a peatland ecosystem in Late Pleistocene North Sea basin began with a birch dominated woodland, impacted by carr vegetation due to paludification and followed by the presence of Chenopodiaceae due to marine inundation (Wolters et al., 2010). The period of succession spanned 1,300 years (Wolters et al., 2010). A high degree of peatland plant community variability results from the highly heterogeneous, irregular, and micro-
ecosystem nature of peatlands (Clymo et al., 1998). It is likely that concurrent sea level-independent paludification occurred in isolated topographic features (e.g. local pools, valleys, streams), impacting local vegetation succession. The vegetation description of Wolters et al. is primarily based on pollen analysis, representative of regional-scale changes. To establish an understanding on the degree of variability between sites, more research is needed to compare the results of the Wolters et al. site with other locations in the North Sea basin.






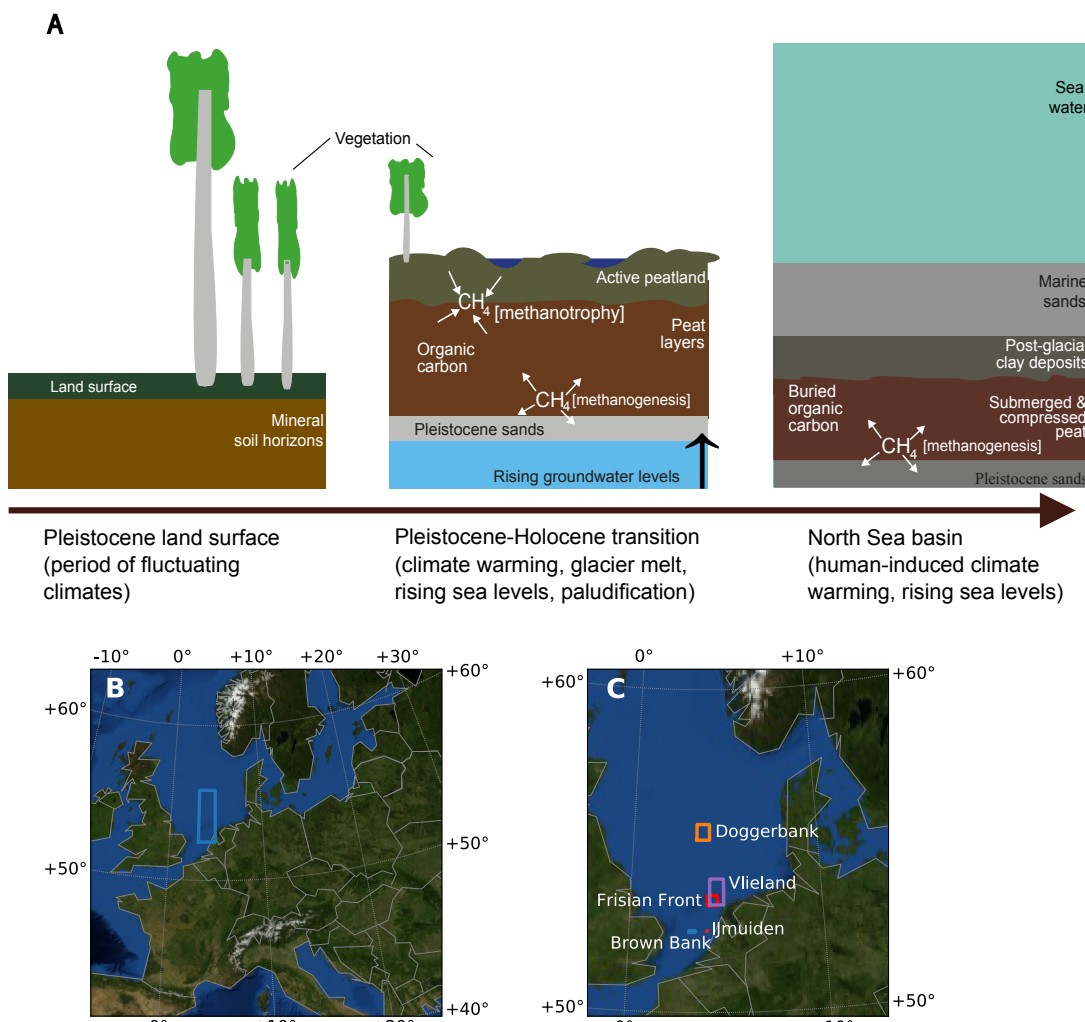

**Figure 1: Peats submerged beneath the North Sea region of study. (A)** Schematic of the evolution of processes that led to the conversion from the Pleistocene land surface to the buried marine peat sediments as they occur today. **(B)** The location of the sampling area within the context of Western Europe. **(C)** The distribution of tsites within this sampling area. B. and C. were created using Python's Basemap module and their background images use NASA's Earth Observatory's, Blue Marble: Next Generation.

Despite extensive efforts to map basal-peats at the global scale in recent decades (Treat et al., 2019; Xu et al., 2018), the task of measuring $CH_4$ stores remains challenging (Dean et al., 2018). Seismic surveys indicate stores of naturally occurring biogenic $CH_4$ within the North Sea basal-peat deposits (Missiaen et al., 2002), but the presence of $CH_4$ in the widespread basal-peat deposits has not been confirmed or quantified by *in situ* observations. Consequently, these deposits are omitted from the global accounting of C and $CH_4$ budgets of marine sediments (Saunois et al., 2020).

In most North Sea surface waters, $CH_4$ concentrations are typically <0.005 µM L$^{-1}$ (Borges et al., 2016; Niemann et al., 2005). However, much higher $CH_4$ concentrations (1.1 µM L$^{-1}$), among the highest in the world, are observed in the southern North Sea water column off the coast of Belgium (Borges et al., 2016). The release of $CH_4$ from



blowout craters linked to gas exploration could contribute to the high $CH_4$ concentrations in the water column in the
North Sea (Schneider von Deimling et al., 2015; Steinle et al., 2016), but the basin-scale impacts are uncertain
(Rehder et al., 1998). The basal-peat deposits that are widespread beneath the seafloor may be an important
source of basin water $CH_4$.

Microbial activity plays a large role in the biological $CH_4$ cycle and is estimated to be responsible for reducing
annual seabed $CH_4$ emissions to the atmosphere by 1–35 Tg $CH_4$ (Saunois et al., 2020) or   8–65 Tg
$CH_4$ (Reeburgh, 2007). In other words, 50-90% of $CH_4$ produced belowground is estimated to have been oxidised
before reaching the atmosphere (Frenzell and Karofeld, 2000). Numerous studies have measured $CH_4$ fluxes from
present-day peatlands (e.g.(Hendriks et al., 2007; Tiemeyer et al., 2016)). Microbial $CH_4$ production is performed
by methanogens that carry out the final steps in the anaerobic degradation of organic matter. Methanogenesis is
countered by the activity of methanotrophic microorganisms that oxidize $CH_4$ to carbon dioxide ($CO_2$) using a variety
of electron acceptors (in 't Zandt et al., 2018). The relative activity of methanotrophic versus methanogenic
microorganisms plays a determining role in $CH_4$ emissions to Earth's atmosphere (Frenzell and Karofeld, 2000).

Microbial surveys of phylogenetic or functional gene markers have shown that bacterial community composition is
generally distinct between different types of ecosystems, e.g. peatlands (Cadillo-Quiroz et al., 2006), estuarine and
marine sediments (Purdy et al., 2002), tundra and permafrost (Ganzert et al., 2007). Marine microbial communities
are highly diverse and include many uncultured phylotypes (Fry et al., 2008). Community composition is often
similar between ecosystems with common environmental parameters (Kim et al., 2018), but there is a lack of
knowledge of the microbial processes contributing to the production of $CH_4$ in submerged peat deposits globally
and in the North Sea in particular.

To provide a better understanding of the *in situ* ecosystem, particularly its role in the $CH_4$ cycle, here we present
the *in situ* $CH_4$ concentrations and sediment organic matter content of submerged basal-peat deposits in the North
Sea basin. Plant macrofossil analysis was performed to determine plant community composition and describe the
habitat available to micro-organisms now and since peat submersion. 16S rRNA gene amplicon sequencing was
performed to determine microbial diversity, and batch incubations were conducted to investigate actual and
potential microbial $CH_4$ cycle activity in the submerged peat deposits

## 2 Methods

### 2.1 Study Area
The study region (Fig. 1b) spans 150 km east to west, bordering the United Kingdom (3°) and the coastal barrier
system of the Netherlands (5°), and 371 km north to south, extending from the latitude of IJmuiden (52°), the
Netherlands, to the latitude of Kiel (56°), Germany. This region includes the Doggerbank, Frisian Front, and the
Brown Bank and ocean depths ranging from 19 m to 60 m.

A total of 34 cores were collected across 20 sites (see Supplementary Table S1 for the location and analysis details
of all cores and sites). The sites were named according to nearby shipwrecks using Emodnet (EMODnet, 2018),
with the exception of Darci's site.

CH4 measurements were performed at all sites. Four sites in the Vlieland (Vittorio, Max Gundelach, Senator
Westphal SW, Westland) and Doggerbank (Dorthea Shallow SW, Dorthea SSW, Dorthea NW, Fredricksborg NE)



regions were chosen for microbial sequencing analysis and microbial activity studies, respectively. Loss On Ignition (LOI) was performed on these eight sites. Two sites were chosen for plant macrofossil analysis (Vittorio and Fredricksborg NE) and radiocarbon dating. Cores were photographed and texturally described at the facilities of TNO-GDN according to protocol (Bosch, 2000) and will be available at www.dinoloket.nl.


### 2.2 Sediment Sampling

Shallow vibrocores (3 - 4.5m) were collected during two separate cruises in June 2017 and July 2018 on board the NIOZ research vessel *Pelagia*. Before the cruise, digital maps of peat occurrences in the southern North Sea were prepared using the DINO digital database of TNO-Geological Survey of The Netherlands (Van Der Meulen et al.,

2013). Based on these maps, areas were selected for geophysical research using sub-bottom profiler and sparker systems. The seismic profiles were directly interpreted on board to identify the presence and depth of basal-peat beds. The seismic data were used to select sampling sites from a range of water depths and latitudes and within the capability of the vibrocorer (within 4 m of the seabed).

At each site, two or three cores were collected. From one core, samples for $CH_4$ were taken as soon as the core was on deck. Before sampling or after (in the case of $CH_4$ sampling only), the cores were cut into 1-m sections. One of the three recovered cores was cut lengthways following recovery. Sediment for molecular analysis was sampled immediately after opening the core sections and frozen at -80°C until further analysis. Subsequently, pore water samples were collected, and sedimentary samples were taken to determine porosity. All sections were

stoppered and sealed at the base and top and stored in a refrigerated container (4°C).

### 2.3 Methane Sampling

The analysis of submerged peats followed standard protocols for headspace sampling and analysis in marine sediments based on (Reeburgh, 2007) Reeburgh and described further elsewhere (Egger et al., 2015, 2017). Peat

was recovered at all sites, except Easting Down, Stormvogel, and Darci's site. The seismic signal at the Easting Down and Stormvogel sites indicated a peat layer. However, the layers were beyond the range of the vibrocorer, and contained a peat-like gyttja (compact organic matter accumulated on the floor of an inland water body), at the Easting Down and Stormvogel sites, respectively.

Prior to coring, the core liner was pre-drilled with 2-cm diameter holes at 10-cm resolution and taped to be gastight. Upon retrieval, the top and bottom of the core were stoppered immediately. Working with speed, custom-made metal syringes were inserted into each taped hole. Ten millilitres of wet sediment was extracted and deposited into a 65-ml glass bottle filled with saturated NaCl solution. Each bottle was sealed with a black rubber stopper and a screw cap and stored upside down.


In the laboratory, the $CH_4$ bottles were prepared for $CH_4$ analysis. Ten millilitres of $N_2$ was injected into each $CH_4$ bottle (with a needle inserted through the septum, allowing excess water to escape) to create a headspace. From this headspace, a subsample was collected with a gas-tight syringe. Methane concentrations were determined after injection into a Thermo Finnigan TRACE™ gas chromatograph (Flame Ionization Detector) at Utrecht

University. The concentrations reported here were calculated using the measured sediment porosity ($cm^3$ per $cm^3$).

### 2.4 Sediment analysis



Sub-samples for loss on ignition (LOI) analysis were taken at the eight sites corresponding to the sites of the molecular analyses. Sub-sampling was performed at 2-cm resolution within peat layers and 10-cm resolution in
non-peat layers. Organic matter (OM) content (in %) was measured using a Leco® TGA701 Thermogravimetric Analyzer at Vrije Universiteit Amsterdam. Dried samples were weighed, burned at 330°C and 550°C, and weighed again to calculate the mass loss (%). Fixed-volume sub-samples (42.39 cm$^3$) were taken to measure volumetric water content, bulk density, and porosity. These samples were saturated with deionized water, weighed, dried at 60°C for five days, and then re-weighed. The volume of the dry mass was measured using an air pycnometer. The
bulk density of the soil material was calculated by dividing the volume of the dried sediment by the dry mass. The total pore space was calculated by subtracting the measured volumes of water from the original fixed sample volume. Total carbon storage was estimated using the following formula (Erkens et al., 2016):

Total C (kg) = $OM_d$ * V * $C_f$

where $OM_d$ is organic matter density (kg m$^{-3}$), V is peat volume (m$^{-3}$), and $C_f$ is an OM-to-carbon conversion factor for peat deposits.

### 2.5 Pore water analysis

Samples for pore water analysis were extracted using 5-cm MOM Rhizon samplers (produced by Rhizosphere
Research Products, Wageningen) at 10-cm resolution and stored at 4°C until further analysis. The samples were acidified with 1% $HNO_3$ and analyzed by inductive coupled plasma-optical emission spectrometry (ICP-OES) for Al, Ca, Fe, K, Mg, Mn, Na, P, S, Si, and Zn (iCap 6300, Thermo Scientific, Waltham, MA) and continuous flow analysis (CFA) for $NO_3^-$, $NH_4^+$, $PO_4^{3-}$, $Na^+$, $K^+$ and $Cl^-$ (Bran+Luebbe Auto Analyzer, SPX Flow, Norderstedt, Germany; Seal Analytical AutoAnalyzer 3, Seal Analytical, Southampton, UK; Table S2.).

### 2.6 Molecular analyses

Four cores in the southern North Sea were selected for 16S rRNA amplicon sequencing, and four cores from the Doggerbank area were selected for microbial activity studies.

### 2.6.1 DNA isolation

Samples for DNA isolation were immediately extracted aseptically upon sampling. Samples were stored at -20°C until further analysis. DNA was extracted in duplicate per sample using the Qiagen DNeasy Power Soil Kit (Qiagen, Venlo, the Netherlands) following the manufacturer's instructions with the following modifications: the initial PowerBead Tube vortex step was carried out using a TissueLyser LT (Qiagen, Venlo, the Netherlands) at 50 Hz
for 10 min, and the primary centrifugation step was increased to 1 min at 10,000 xg. DNA was eluted with 2×30 μl of sterile Milli-Q incubated for 2 min at room temperature prior to centrifugation. The second elution centrifugation step was carried out for 1 min at 10,000 xg. DNA quality was assessed by agarose gel electrophoresis, spectrophotometrically using a NanoDrop 1000 (Invitrogen, Thermo Fisher, Carlsbad, CA, USA) and fluorometrically using the Qubit dsDNA HS Assay Kit (Invitrogen, Thermo Fisher, Carlsbad, CA, USA) according to
the manufacturer's instructions. Duplicate samples with the highest yield and quality were selected for downstream application.

### 2.6.2 Amplicon sequencing and analysis

For DNA purification, the QIAquick PCR Purification Kit was used (Qiagen, Venlo, the Netherlands). For DNA
amplification, a 2-step amplicon sequencing protocol was used. In the first step, the V3-V4 region of the bacterial



16S rRNA gene was amplified using the universal primers Bac 341F (5′-CCTACGGGNGGCWGCAG-3′) (Herlemann et al., 2011) and Bac785R (5′-GACTACHVGGGTATCTAATCC-3′) (Klindworth et al., 2013) for 30 cycles. Archaeal 16S rRNA genes were amplified with the universal archaeal primers Arch349F (5′-GYGCASCAGKCGMGAAW-3′) (Takai and Horikoshi, 2000) and Arch806R (5′-GGACTACVSGGGTATCTAAT-3′)

(Takai and Horikoshi, 2000) for 30 cycles. All primers were purchased from Biolegio (Biolegio B.V., Nijmegen, the Netherlands).

The following cycling parameters were used for polymerase chain reaction (PCR): initial denaturation for 10 min at 98°C; 25/30 cycles of denaturation for 1 min at 95°C, annealing for 1 min at 60°C, and elongation for 2 min at 72°C;

and a final elongation step for 10 min at 72°C. PCR products were purified using the QIAquick PCR Purification Kit (Qiagen, Venlo, the Netherlands) in two elution steps. A 20-µl aliquot of 55°C Milli-Q water was added to the spin column and incubated for 2 min prior to centrifugation. Next, the eluate was added to the spin column, incubated at 55°C for 2 min, and centrifuged again as described in the manual. The purified PCR products were used in a second 10-cycle nested PCR performed with IonTorrent adapters using the PCR protocol described above. After

purification with the QIAquick PCR Purification Kit as described earlier, the PCR products were used for library preparation and sequencing steps according to the manufacturer's instructions (Life Technologies, Carlsbad CA, United States).

Samples were sequenced on an Ion 318 Chip Kit v2 (Thermo Fisher, Waltham, MA, USA). Amplicon sequences

were quality checked for chimeras and clustered into OTUs with a 97% identity cut-off value using the 454 SOP (http://www.mothur.org/) (Schloss et al., 2009) with IonTorrent modified protocols. Chimeras were checked with the Uchime algorithm (Edgar et al., 2011), singletons were removed (Fig. S1). Taxonomy was assigned against the SILVA nr v132 database using the 'mothur' taxonomy assigner (Schloss et al., 2009). Data visualization was performed using the 'vegan' package in 'r' (Oksanen et al., 2019). All alpha diversity indices were calculated with

the OTU-based alpha diversity analysis tool summary.single() of 'mothur'. Non-metric dimensional scaling (NMDS) plots were prepared in 'r' using the 'vegan' and 'MASS' packages after pre-filtering of non-abundant OTUs (Venables and Ripley, 2002). OTUs with a sum of ≤ 1 per sequencing dataset were removed from the OTU table. NMDS ordination was performed with the metaMDS() function of 'vegan'. Data were processed by square root transformation and Wisconsin double standardization.


### 2.6.3 PCR quantification, cloning and qPCR

16S rRNA gene copy numbers were quantified with the archaeal and bacterial primers described above, except that for bacterial quantification, the primer Bac806R (5′-GGACTACHVGGGTWTCTAAT-3′) (Caporaso et al., 2012) was used. Quality and size checks were performed by agarose gel electrophoresis. All qPCR reactions were

performed using PerfeCTA Quanta master mix (Quanta Bio, Beverly, MA) and 96-well optical PCR plates (Bio-Rad Laboratories B.V., Veenendaal, the Netherlands) with optical adhesive covers (Applied Biosystems, Foster City, CA). All reactions were performed on a C1000 Touch thermal cycler equipped with a CFX96 Touch™ Real-Time PCR Detection System (Bio-Rad Laboratories B.V., Veenendaal, the Netherlands); a maximum of 1 ng of DNA template was used per reaction. Negative controls were prepared for each run by replacing the template with sterile

Milli-Q water. Standard curves were constructed with a 10-fold serial dilution of a quantified copy number of pGEM®-T Easy plasmids containing inserted Illumina primer PCR fragments of the archaeal and bacterial 16S rRNA genes (Promega, Madison, WI). All qPCR data were analyzed using Bio-Rad CFX Manager version 3.0 (Bio-Rad Laboratories B.V., Veenendaal, the Netherlands).



### 2.7 Activity studies

#### 2.7.1 Incubation experiments

Sediment cores for the activity study were drilled on 27 and 28 June 2018 and stored for 10 months at 4°C until further processing. Material was taken aseptically from the carbon-rich (dark black/brown) peat layer and stored in sterile 50-ml falcon tubes kept on ice at 4°C during transport. A total sediment slurry volume of 1.5 l was obtained by mixing 750 g (0.5 l volume) of peat with artificial sea water (0.546 M $Cl^-$, 0.469 M $Na^+$, 0.0528 M $Mg^{2+}$, 0.0282 $SO_4^{2-}$, 0.0103 M $Ca^{2+}$, 0.0102 M $K^+$, 0.0012 M $CO_3^{2-}$, 0.000844 M $Br^-$, 0.000091 M $Sr^{2+}$, 0.000416 M $B^-$, 0.00935 M $NH_3^+$, 0.00367 M $PO_4^{3-}$) (Dickson, A. G. & Goyet 1994) amended with 1 ml/l 1000x trace element solution SL-10 with 24 mg $l^{-1}$ $CeCl_3 \cdot 7H_2O$, 30 mg $l^{-1}$ $Na_2SeO_3 \cdot 5H_2O$ and 40 mg $l^{-1}$ $Na_2WO_4 \cdot 2H_2O$ (DSMZ) and adjusted to pH 7.0. Under continuous mixing, 50-ml sludge aliquots were transferred to 120-ml sterile glass serum bottles. The bottles were sealed with airtight butyl rubber stoppers and capped with open-top aluminium crimp caps. All incubations were carried out in triplicate per condition.

Methanogenic incubations were carried out anoxically with acetate (20 mM), $H_2/CO_2$ (20 mM $H_2$ with 20% $CO_2$ in headspace), $H_2$/methanol (10 mM $H_2$, 10 mM MeOH), and trimethylamine (20 mM). For methoxydotrophic methanogenesis, incubations were started with methoxyphenol (3 mM) and trimethylbenzoate (3 mM). For sulfate-dependent methanotrophy, samples were incubated with 28.2 mM sulfate, the concentration present in the artificial seawater, and 5% (~2 mM) $^{13}C-CH_4$. The anoxic control incubations were unamended. Anoxic conditions were created by three 15-min cycles of vacuuming and subsequent gassing for 3 min with 1 bar overpressure. The overpressure was removed before starting the incubations. The gas mixture contained 80% $N_2$ and 20% $CO_2$ except for the incubations for hydrogen-dependent methylotrophic methanogenesis, which were gassed with 100% $N_2$. To remove trace oxygen, 0.5 ml of 150 g/l L-cysteine-HCl and 0.5 ml of 150 g/l $Na_2S$ were added. To inhibit excessive growth of sulfate-reducing bacteria, a sterile sodium molybdate solution was added at a final concentration of 1.5 mM to all incubations with $H_2$ (Banat Nedwell and Balba, 1983). A new dose of 10 mM $H_2$ was added to the $H_2/CO_2$ incubations at 30 and 49 days of incubation and to the $H_2$/methanol incubations at 35 and 49 days of incubation. A second dose of 10 mM MeOH was added to the $H_2$/methanol incubations at 63 days of incubation.

For aerobic methanotrophic incubations, air was used as the headspace and amended with 10 mM $CH_4$. Oxic control incubations contained only air. All substrate concentrations were calculated based on a liquid volume of 50 ml and assuming that all of the substrate dissolved over time.

For substrate consumption rates, the per $cm^3$ substrate conversions were calculated by dividing the total substrate conversion numbers by 16.67 $cm^3$, which corresponds to the quantity of compacted peat sediment inoculated per batch incubation.

#### 2.7.2 Substrate and product analysis

Gas samples (50 µl) were withdrawn with a gas-tight glass syringe (Hamilton, Reno, NE) and injected into an HP 5890 gas chromatograph (Hewlett Packard, Palo Alto, CA) equipped with a Porapaq Q 100/120 mesh (Sigma Aldrich, Saint Louis, MI) and a flame ionization detector (FID) for $CH_4$ detection and a thermal conductivity detector (TCD) for measuring $H_2$, $CH_4$ and $CO_2$ simultaneously using $N_2$ as the carrier gas. An Agilent 6890 series gas





chromatograph coupled to a mass spectrometer (Agilent, Santa Clara, CA) equipped with a Porapak Q column heated at 80°C with He as the carrier gas was used for measurements of $^{13}CO_2$, $^{13}CH_4$ and $O_2$.

**2.8 Plant macrofossil analysis**

Two sites, the Max Gundelach site and the Fredricksborg NE site, were selected for plant macrofossil analysis. The Max Gundelach site is in the southern North Sea near the coast of the Netherlands (4°51.07'E, 53°20.09'N), and the Fredricksborg NE site is in the Doggerland region (3°26.42'E, 55°49.48'N).

The Max Gundelach site was analysed with low sample resolution but high taxonomic resolution, showing the main peat components as well as an overview of the less abundant taxa. The Fredericksborg NE site was analysed with high sample resolution but low taxonomic resolution, showing only the main peat components.

From the Max Gundelach core, 8 samples (slices with a thickness of 1 cm or, in two cases, 2 cm and a volume
ranging from 8 to 11 cm³) for plant macrofossils were taken every 10 cm. From the Fredricksborg NE core, 15 subsamples were taken with a resolution ranging from 1 to 4 cm and volumes ranging from 3 to 8 ml. The samples were heated near the boiling point in 5% NaOH solution and then gently washed through a 150-μm mesh sieve with tap water. After sieving, the plant macrofossils were stored in a known volume of water. The sample material was systematically examined at 15 to 40X magnification using a stereomicroscope.


The main peat components (monocot epidermis, brown mosses, *Sphagnum* spp.) of both cores were quantified based on the quadrat and leaf Count (QLC) method (Barber et al., 1994, 2003) using 15 averaged quadrat (1 x 1 cm) counts under low power (X10) magnification using a 10 x 10 square grid graticule. The main peat components were expressed as percentages (%). The complete samples were scanned for quantification of the less abundant
macrofossils, in case of the Max Gundelach core, and seeds, fruits, leaves and fragments of mosses were picked out, counted and expressed as concentrations per unit of volume. Rare taxa are reported as presence. Preservation of the peat deposits was estimated qualitatively during analysis based on the preservation of the macro fossils: poorly preserved (+), intermediately preserved (++) and well preserved (+++). The diagram was constructed with Tilia Version 1.7.16 (Grimm, 2004).


**2.9 Radiocarbon-dating**

For radiocarbon-dating purposes, the top and bottom 1 cm of the peat layers were sieved and searched for autochthonous terrestrial plant macrofossils or, in the absence of such fossils, charcoal (Hijma and Cohen, 2010). If a 1-cm-thick section did not contain enough macrofossils, material from the subsequent cm was added. The
selected macrofossils were sent to the Centre for Isotope Research (Groningen, the Netherlands) for AMS-radiocarbon dating. All radiocarbon ages were calibrated using OxCal 4.3 software (Bronk Ramsey, 2009) with the INTCAL13-curve (Reimer et al., 2013).

**3 Results**


**3.1 Lithographic features of the sampling sites**

The localized nature of this landscape is apparent in the lithographic differences observed between and within sites (Fig. 2). The peat deposits in this area lie upon Pleistocene sands and are capped by either shallow marine clay or sands. Pleistocene sands lie 2-4 m below the sea floor (mbsf) in the southern North Sea and 1-3 mbsf in the



Doggerbank region and are capped by 80-120 cm and 10-30 cm basal-peat layers in the southern North Sea and Doggerbank regions, respectively. The basal-peat developed due to rising groundwater as a result of the postglacial sea-level rise and was quickly capped by rapidly deposited clays and subsequent sand deposits in most instances or directly capped by sand in others.

Non-erosive contact transitions exist between the peat layer and both the immediate upper and lower sedimentary layers at 17 sites, indicating that no widespread extreme event took place in the period directly prior or subsequent to the Pleistocene-Holocene transition. In most cases, the peat is covered by a clay layer that formed under lagoonal, low-energy conditions and rests conformably on the peat. However, some cores do show erosional contacts (i.e. Fredricksborg NE (Fig. 2h), Fredricksborg NW, and Dorthea NW sites) at the top of the peat beds

that can be associated with marine transgression of the area, likely related to waves or tidal currents. The clay sediment directly above the peat layer at the Vittorio site is approximately 1 m thick and the thickest clay deposit among all sites. The nearby Max Gundelach site has clay deposits capping the peat that are ±35 cm thick. At the other sites, the clay deposits capping the peat layer range from 5 cm to 50 cm.

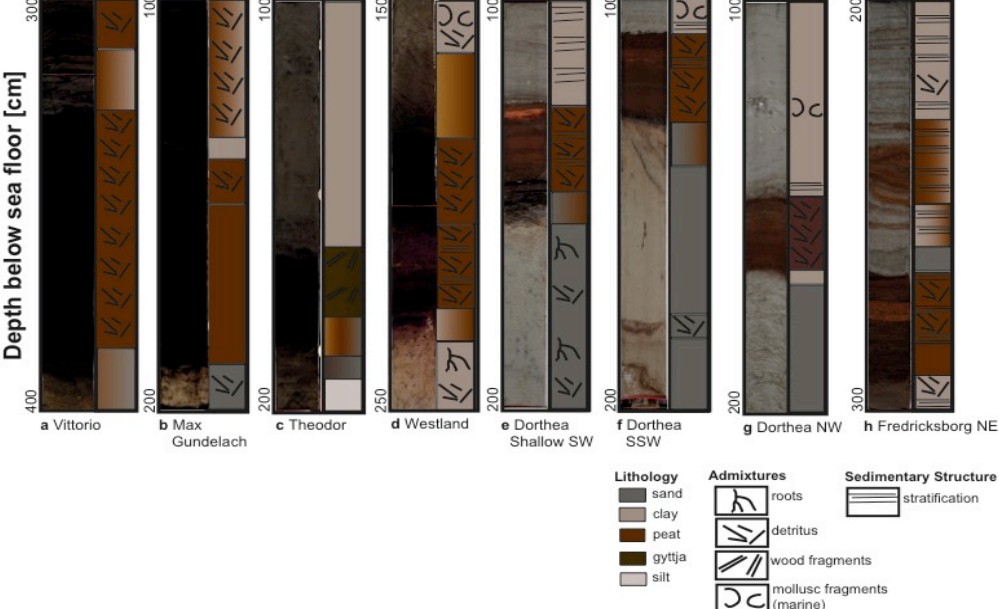

**Figure 2: Photographs and stratigraphy of key sites. (A-D)** The four sites from which sediments were used to perform 16S rRNA gene-based diversity analysis. These sites lie within the mid and southern North Sea. **(E-F)** The four sites from which sediment was used to study microbial activity. These sites originate from the Doggerbank area. Note the varying y-axes.

**3.2 Low CH$_4$ concentrations are widespread, high local concentrations occur**

The average CH$_4$ concentration of the sediment pore waters was 2.1 µmol L$^{-1}$, with a maximum concentration of 32.8 µmol L$^{-1}$ (Fig. 3). Ten sites had CH$_4$ concentrations lower than the study average, i.e., <2 µmol L$^{-1}$: TX24, Theodor, Mahren S, Easting Down, Leda, Dorthea Shallow SW, Dorthea NNW, Dorthea NW, Fredricksborg NE, and Easting Down. Ten sites had CH$_4$ concentrations similar to or above the study average, i.e., >2 µmol L$^{-1}$:

Vittorio, Max Gundelach, U21, Senator Westphal, Westland, Fredricksborg NW, Stormvogel, Dorthea Deep SW, Darci's Site and Dorthea Shallow SW. One of the two cores retrieved from the Dorthea Shallow SW site had low



CH₄ concentrations, while the second had high CH₄ concentrations. Overall, we found approximately equal numbers of sites with high and low CH₄ concentrations, indicating a high degree of spatial variability.

The highest concentration of CH₄ was observed at the Vittorio site, at the latitude of Vlieland. The Vittorio site had the second thickest peat layer in this study, but the thickness of the peat layer does not appear to play a determining role in CH₄ concentrations, as both thick and thin peat layers harboured both high and low CH₄ concentrations.

Four sites (Fig. 2a-d) representing both high and low CH₄ concentrations were chosen for 16S rRNA gene-based
sequencing to unravel the microbial community structure. A second cluster of four sites (Fig. 2e-f) was chosen from the Doggerbank area to investigate the role and potential of *in situ* microbial communities in CH₄ cycling.

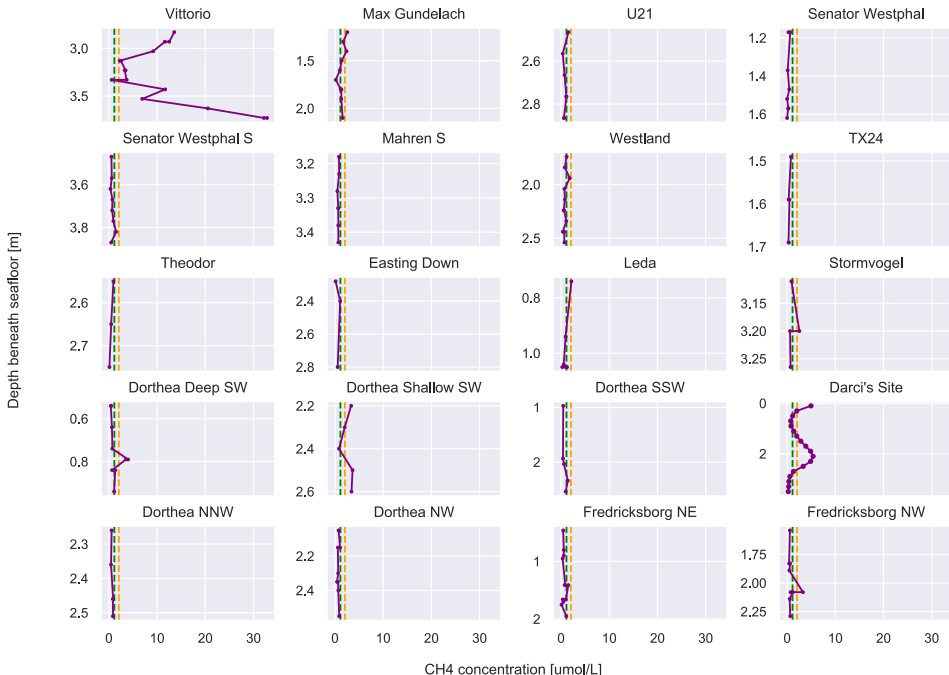

**Figure 3: Methane profiles**. Depth profiles of methane concentrations at all sites in µmol L⁻¹. The yellow line indicates the average methane concentration across all measurements. The green line indicates the average
methane concentrations of seawater measured in the same area (Borges et al., 2016).

### 3.3 Peat developed due to rising groundwater during climate warming

Radiocarbon dating (Table 1) indicated that peat formation began 13,680 calibrated years before present (cal yr BP) at the Fredricksborg NE site (55°49.49', 3°26.40', Doggerbank region) and 11,760 cal yr BP at the Max
Gundelach site (53°20.4', 4°51.6', nearby Vlieland). Peat formation ceased 12,880 cal yr BP at the Fredricksborg NE site and 8,290 cal yr BP at the Max Gundelach site.

At the Max Gundelach site in the Vlieland region, radiocarbon-dating revealed that active peat formation prevailed for approximately 2,000-3,000 years longer than at the Fredricksborg NE site in the Doggerbank region. The period
of active peat formation depended on the ability of peat formation to keep up with the rising groundwater table and on hydrological conditions; e.g., peat formation commenced earlier in areas with a less permeable substrate than in areas with a sandy substrate. For plant macrofossil analysis, one of the most southern sites, Max Gundelach,





and one of the most northern sites, Fredricksborg NE, were chosen as representative of the local onset and termination of North Sea peat development, respectively.

**Table 1. Radiocarbon dates.** $^{14}$C dates of the Max Gundelach site in the Vlieland area and the Fredricksborg NE site in the Doggerbank area, the two sites where plant macrofossil analysis was performed.

| Site name | Depth below seafloor (cm) | Lab number | Dated material | $^{14}$C age $^{(BP)}$ | Calibrated age (95% min & max age range) |
|---|---|---|---|---|---|
| Max Gundelach | 104-106 | GrM-17947 | Charcoal | 7475 ± 35 | 8290 (8,190-8,380) |
| Max Gundelach | 106-108 | GrM-17751 | Cladium mariscus 53 | 7540 ± 35 | 8360 (8,220-8,420) |
| Max Gundelach | 123-125 | GrM-18853 | Carex plat 3, 1/3 Carex driehoekig, Betula pubescens/pendula 1 | 7890 ± 40 | 8720 (8,590-8,980) |
| Max Gundelach | 188-190 | GrM-17752 | Carex sp. 22; Typha sp. 4 | 10120 ± 35 | 11760 (11,500-12,010) |
| Fredricksborg NE | 265-267 | GrM-19239 | Cyperaceae 25 | 11020 ± 40 | 12880 (12,740-13,010) |
| Fredricksborg NE | 289-291 | GrM-19287 | Carex sect. Acutae 35 | 11885 ± 40 | 13680 (13,570-13,780) |

**3.4 Plant macrofossil analysis and local vegetation history**

The peat layer of the Max Gundelach site is approximately 85 cm thick, and radiocarbon dating revealed that an active peatland persisted for 3,470 years between 11,760 and 8,290 cal yr BP. This is a far longer period than what occurred at the Fredricksborg NE site, where the peat layer is only 10 cm thick and radiocarbon dating indicated that a peatland was active during an earlier and shorter 800-year period between 13,680 and 12,880 cal yr BP.

Plant macrofossil analysis denoted that peat accumulation occurred through paludification due to a rising water table at both the older northern and younger southern sites.

**3.4.1 Local vegetation succession in the southern North Sea**

At the Max Gundelach site, wet terrestrial vegetation was present at the start of peat accumulation 11,760 years

BP, with the presence of *Carex* spp. (Fig. 4a). A certain degree of open water was present, as remains of invertebrates (*Chironomid* head capsules, Cladocera) together with Characeae oospores were found. These green algae are characteristic of lake waters in pioneer conditions with inputs of minerogenic material (Mauquoy and Van Geel, 2007) and therefore indicative of eutrophic conditions. At 180 cm beneath the seabed, Cladocera resting eggs were found, suggesting harsher conditions for these invertebrates. All other open-water taxa disappeared.


From a depth of 170 cm onward, the environment became nutrient poor, as evidenced by the dominance of *Sphagnum magellanicum*. *S. magellanicum* is an important contributor to ombrotrophic peat bogs with a constant water table (Siebel and During, 2006). Remains of woody plants, in the form of leaf scars, and charcoal were also present at this depth. This indicates the presence of vascular plants during peatland growth.


*S. magellanicum* declined and the brown moss *Tomentypnum nitens*, a species no longer existing in the Netherlands*,* became the main peat building component at 160 cm depth. *T. nitens* is an indicator species of mineral-rich fens, highlighting a change from nutrient-poor to nutrient-rich conditions. The presence of *T. nitens* indicates that calcium and nutrient-rich groundwater were seeping into the terrestrial environment (Bohncke et al.,

1984; van Geel et al., 2020; Hedenäs and Kooijman, 1996).

From 150 cm depth onward, *T. nitens* was replaced by *Warnstorfia sarmentosa* and *Drepanocladus* sp. Both species are brown mosses, further indicating a transition to wet mesotrophic conditions. *Carex* sp. (sedges) rootlets were found in the top of the peat sequence.





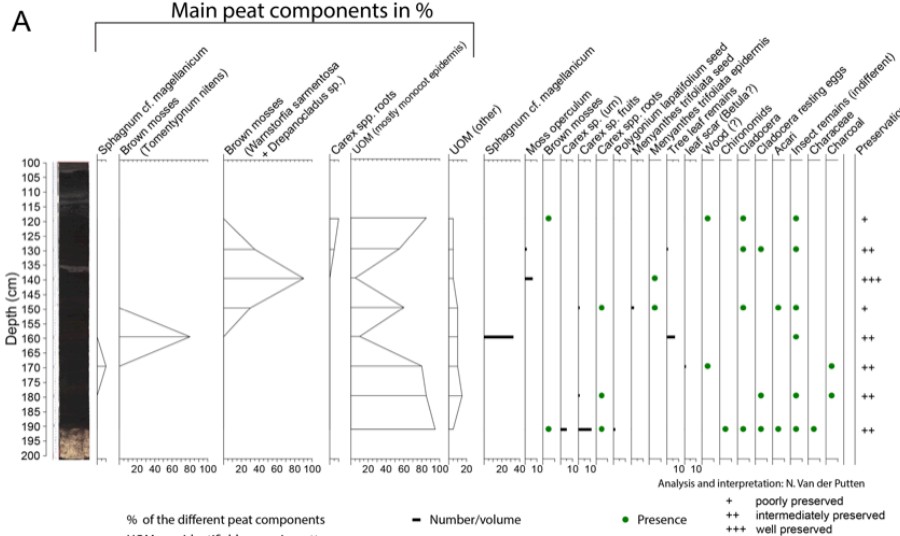

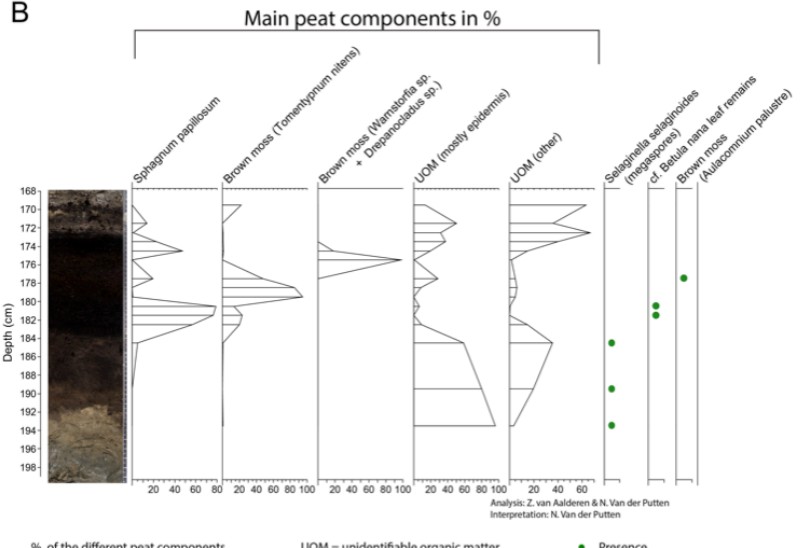

**Figure 4: Macrofossil diagrams** of the **(A)** Max Gundelach site. Preservation of the plant remains are qualitatively estimated and expressed using a three-step scale: + (good), ++ (very good) and +++ (excellent); and **(B)** Fredericksborg NE site against depth and core photograph. The main peat components of both sites are quantified using the quadrat and leaf Count (QLC) method (see methods). They are expressed as percentages (%) and are shown as hollow curves, with the lines indicating the depth of the samples. The complete sample was screened for additional less abundant taxa which are expressed as concentrations (number of remains per unit of volume) and shown as black bars. Rare taxa are shown as presence with green dots.

### 3.4.2 Local vegetation succession in the mid North Sea at Doggerbank

In the lower part of the Fredricksborg NE core (Fig. 4b), from 191 cm onward, a change from nearly purely minerogenic substrate (LOI550 is c. 1%) to slightly higher LOI550 values, varying around 6% (Fig. S2), points to the presence of a sparse pioneer vegetation. In the three lower samples, megaspores of *Selaginella selaginoides* were found. *S. selaginoides* is a heliophilous (needing/tolerating a high level of direct sunlight) circumpolar Boreal-montane species growing in damp neutral to alkaline conditions, including dune-slacks, fens, flushes, mires and



short upland grassland (Tobolski and Ammann, 2000). In Northern Scandinavia it occurs in mires, at lake margins and damp heath meadows (Bjune et al., 2004). Peat formation through paludification started at 183 cm depth, evidenced by a sudden increase (to 75%) of organic material burnt at 550°C.

The plant macrofossil content of the peat deposits in Fredricksborg NE (Fig. 4b) is dominated by bryophytes, *Sphagnum* as well asbrown mosses. Peat accumulation began with *Sphagnum papillosum*, quickly followed by the brown moss *Tomentypnum nitens* and subsequently by the brown mosses *Warnstorfia* sp. and *Drepanocladus* sp. *Sphagnum papillosum* is a typical moss of acid raised bog but in the Netherlands it also occurs in fenland areas as well as sand regions, including dune-slacks i.e. on the Wadden Islands (Bryologische en Lichenologische

Werkgroep, 2015).

**3.4.3 Local vegetation succession is analogous across sites**

It is striking that the same three-step bryophyte dominated sequence of *Sphagnum-Tomentypnum nitens-Warnstorfia/Drepanocladus* occurs in both geographically as well as temporally different sites. However, in contrast

to the sequence of the Max Gundelach site, where *Sphagnum magellanicum* is present only at the start of the sequence, *Sphagnum papillosum* is present throughout the peat deposits at the Fredricksborg NE site. In general, plant remains are better preserved in the layers dominated by *Sphagnum* spp. than in those dominated by brown mosses.

**3.5 Estimating CH$_4$ storage, organic matter and CO$_2$ equivalents**

The study area (116 km by 372 km, Fig 1b) spans a surface area of 43,158 km$^2$, an area larger than the land surface of the Netherlands (41,865 km$^2$). Based on the average peat thickness of 0.29 m (minimum: 0.07 m, maximum: 0.88 m), the estimated volume of peat is 12.4 km$^3$ (min: 3.0, max: 38.0, km$^3$). Multiplying this estimated volume by the average observed CH$_4$ concentration (2.14 µmol L$^{-1}$), we estimate that 0.411 Tg CH$_4$ (min: 0.100,

max: 1.256, Tg CH$_4$) is present in the study area.

Carbon storage and its CO$_2$ equivalent were calculated using the estimated peat volume of 12.4 km$^3$ and 103 kg m$^{-3}$, the average organic matter density of compressed peat in the Netherlands (Erkens et al., 2016). This volume of peat is estimated to hold 740.8 Tg C (min: 180.4, max: 2,270.1), assuming the convention that dry peat biomass

has carbon concentration 0.5 g C g$^{-1}$ (Gorham, 1991; Heijmans et al., 2008). This is equivalent to 2,716.2 Tg CO$_2$ (minimum: 661.5, maximum: 8,323.8), if released into the atmosphere, assuming a conversion of soil C to CO$_2$ of 1.00:3.67 (Van den Bos, 2003).

**3.6 Variations in organic matter between local environments**

Sediment OM content was highest in the peat-containing layer at all sites (Fig. S2). The percentage of OM burnt at 550°C (LOI550) was greater than the percentage of OM burnt at 330°C (LOI330) in all cores. LOI330 showed greater variation with depth than LOI550 at all sites. The LOI results for the Fredricksborg NE site were distinctive because LOI330 and LOI550 were almost equal.

Scatter plots show that OM burnt at 330°C and 550°C follow comparable trends (i.e., more burnt OM at 330°C corresponded to more burnt OM at 550°C; (Fig. S2)). However, in general, there was a ceiling of not more than 50% OM burnt when exposed to temperatures of 330°C. Proportions of OM burnt at 330 °C was highest for the Vittorio and Fredricksborg NE sites. The maximum OM burnt (80%) at 550°C occurred at all sites except the



Theodor site. The thickness of the peat layer at the Theodor site is thin, 8 cm, which is thinner than the mean peat-
layer thickness of all sites (0.29 m).

### 3.7 Methanogenic archaea actively perform methylotrophic methanogenesis

To investigate the potential of the *in situ* microbial community for $CH_4$ cycling (schematic of process, Fig. 1a), batch incubations were prepared with an anoxic slurry of artificial seawater and freshly collected peat sediment and
amended with a range of methanogenic substrates. Pore water analysis indicated that the peat layers were converted into a marine ecosystem. The peat deposits at Dorthea Shallow SW, Doothea SSW, Dorthea NW, and Fredricksborg NE (showed active $CH_4$ production upon incubation, with a strong increase in rates of production upon methylated substrate amendment (Fig. 5). Molecular analysis showed that both methanogens and methanotrophs were present at all four assessed sites: Westland, Senator Westphal, Max Gundelach and Vittorio
(Fig. 6).

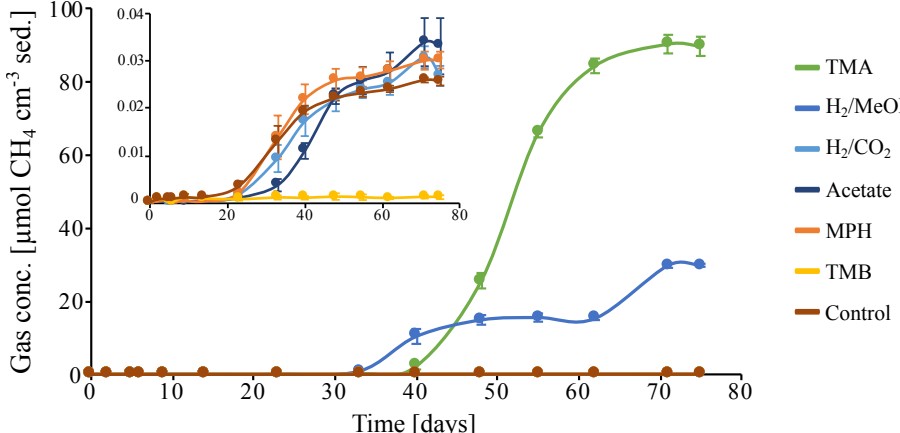

**Figure 5: Methane production in batch incubation assays of peat sediment slurries.** Methane contents are given as $CH_4$ produced per $cm^3$ of original peat sediment over the course of 75 days. Substrates: trimethylamine (TMA), hydrogen and methanol ($H_2$/MeOH), hydrogen and $CO_2$ ($H_2$/$CO_2$), acetate, methoxyphenol (MPH),
trimethoxybenzoate (TMB) and an anaerobic control incubation without substrate amendment (Control). Data points represent the average of triplicate measurements on triplicate incubations. Error bars indicate the standard deviation of the mean. The insert depicts a zoom in on the $CH_4$ concentrations excluding the incubations on TMA and $H_2$/MeOH.

Methane production in the unamended control incubation was very low, indicating that most, if not all, of the labile organic matter fraction of the peat sediments has already been mineralized. Methane accumulation was observed subsequent to the addition of methylated substrates (Fig. 5) after a lag phase of two weeks, indicating that the $CH_4$-producing microbial community could be quickly metabolically revived. In the incubation with $H_2$ and MeOH, $CH_4$ production was solely linked to MeOH, which was confirmed upon amendment with MeOH after depletion of $H_2$.


Amendment with hydrogen and $CO_2$ ($H_2$/$CO_2$) and acetate, two common substrates for methanogenic archaea, did not induce $CH_4$ production within the period of incubation (60 days). Even though no methanogenesis was observed, the concentration of $H_2$/$CO_2$ changed. This may be indicative of competition for substrates, likely by sulfate-reducing microorganisms facilitated by abundant sulfate supplies that penetrate up to meters deep into the
sediment in marine environments (Jorgensen, 1983) or, in this case, incubations with ample supplies of sulfate. Amendment with methoxyphenol (MF) and trimethoxybenzoate (TMB), substrates used by methoxydotrophic methanogens, did not induce $CH_4$ production, and a TMB concentration of 3 mM appeared to be inhibitory to the





methanogenic community. Neither aerobic nor anaerobic methanotrophic activity was observed, indicating the absence of an *in situ* biological $CH_4$ filter (Fig. S3 & Fig. S5).

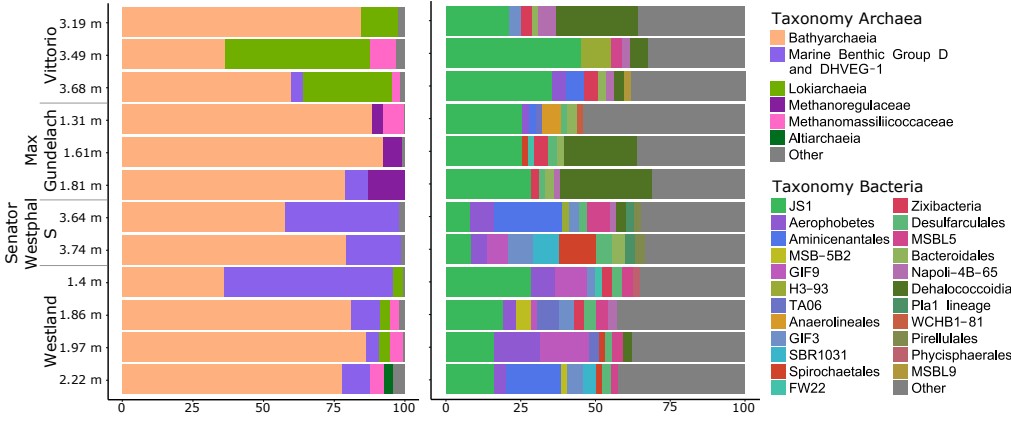


**Figure 6: Phylogenetic classification of amplified archaeal (left) and bacterial (right) 16S rRNA genes** found at the Vittorio, Max Gundelach, Westland, and Theodor sites. The Y-axis values indicate the depth below sea level (dbsl). The maximum taxonomy depth is on family level. Taxonomic groups with < 2% abundance are grouped in 'Other'.


### 3.8 Microbial community composition

*16S rRNA gene quantification shows dominance of archaea over bacteria in all cores*

Archaeal and bacterial abundances in each core section were assessed by quantitative PCR. In all cores, archaea were more abundant than bacteria (Table S3 and Fig. S4). Cores 6 and 7 had archaea-to-bacteria ratios of 7.0 and

9.0, whereas cores 17 and 26 had ratios of 55.1 and 43.9, respectively. Archaeal 16S rRNA gene copy numbers ranged from 1.3 to 8.1 x $10^7$, while bacterial 16S rRNA gene copy numbers ranged from 1.7 to 3.2 x $10^6$.

### 3.8.1 Dominance of Bathyarchaeia and prevalence of methanogenic archaea

Bathyarchaeia dominated the archaeal communities at all locations, with an average relative abundance of 71%

(range 35.9-92.2%). The relative abundance of Bathyarchaeia was highest at the Max Gundelach site, with an average of 86% of the archaeal 16S rRNA gene reads. The phylum Bathyarchaeia is a potentially metabolically diverse microbial group that is found in a wide range of organic-rich environments, including deep sea and freshwater sediments (Evans et al., 2015). Among the four sites for which DNA sequencing was performed, methanogenic archaea were detected at Vittorio, Max Gundelach and Westland but not Senator Westphal S (Fig.

6 & Fig. S4). Methanogenic archaeal species belonging to *Methanomassiliicoccaceae* were detected in these three cores, whereas *Methanoregulaceae* were only observed at Max Gundelach. The Max Gundelach site contained the highest relative abundance of methanogenic archaea, with an average of 10.3%, compared to averages of 3.9% and 3.0% at the Vittorio and Westland sites, respectively.

Lokiarchaea were most abundant at the Vittorio site (32.2%) and were present at only low abundance at the Westland site (2.8%) and below the 2% threshold at the other sites (Fig. 6). Marine Benthic Group D and DHVEG-1 were more abundant at the Senator Westphal S and Westland sites (30.3% and 21.2%, respectively) and were present only at low abundance at the Vittorio and Max Gundelach sites (1.4% and 2.7%, respectively). Genomic analysis of the Asgard candidate phylum "Candidatus Lokiarchaeota" has indicated potential for an acetogenic

lifestyle, hydrogen dependency and mixotrophic potential (Sousa et al., 2016; Spang et al., 2019).





### 3.8.2 Diverse bacterial communities dominated by candidate phylum JS1

Candidate phylum JS1 dominated the bacterial communities, with an average relative abundance of 22.9% (Fig. 6). The highest relative abundances, 33.7% and 26.3%, were observed at the Vittorio and Max Gundelach sites,

respectively. Dehalococcoidia were mainly observed at the Vittorio and Max Gundelach sites, with respective abundances of 12.3% and 18.4%. At the Senator Westphal S and Westland sites, the abundances of Dehalococcoidia were low, with averages of 1.6% and 0.8%, respectively. The JS1 lineage is a subgroup of the candidate phylum Atribacteria (Nobu et al., 2016). Metabolic reconstructions have indicated the potential of JS1 bacteria for fermentative metabolism and syntrophic acetate oxidation (Lee et al., 2018). Aerophobetes and GIF9

phylum bacteria were more characteristic of the Senator Westphal S (6.7%) and Westland (8.1%) sites and were present only at low abundance at the Vittorio (1.6%) and Max Gundelach (0.8%) sites. Candidate GIF9 bacteria were only detected at the Senator Westphal S (3.4%) and Westland (7.2%) sites. In addition, MSB.5B2, TA06, SBR1031, Pla_1 lineage (Senator Westphal S site only), Pirellulales (Senator Westphal S site only) and Phycisphaerales (Westland site only) were unique to specific cores.


### 3.8.3 Archaeal and bacterial diversity analyses

Archaeal species diversity was greater at the Senator Westphal S (Simpson: 0.24; Shannon: 2.13) and Westland (Simpson: 0.19; Shannon: 2.85) sites than at the Vittorio (Simpson: 0.21; Shannon: 2.28) and Max Gundelach (Simpson: 0.23; Shannon: 1.96) sites (Fig. S6a). The archaeal community structure was similar among the cores,

as supported by non-metric multidimensional scaling (Fig. S7).

The high microbial diversity of these peat sediments was reflected in the alpha diversity indices. Compared to inundated mangrove peat soils, the bacterial alpha diversity in the North Sea peat sediments sampled in the present study was higher (Shannon diversity of up to 5 vs 2.81, Fig. S6b.) (Chambers et al., 2016). The indicators of

diversity observed here are comparable to or higher than those observed in tropical peat swamp forests in Thailand (Shannon diversity of 5.07) and Indonesia (2.0-2.5), but the largest estimated Chao1 index was much higher (1,054 for Thailand peat vs 1,500-4,500 observed in our study) (Chambers et al., 2016; Kwon et al., 2016).

### 4.0 Discussion


### 4.1 A newly measured CH$_4$ store

This study presents geochemical conditions, vegetation composition, microbial diversity and metabolic potential in the context of the CH$_4$ cycle. Compared to other studies that have measured CH$_4$ concentrations in marine sediments, the geographic expanse of this study is large (Egger et al., 2016, 2017; Niemann et al., 2005; Steinle

et al., 2016). The broad distribution of the sample locations (study area in Fig. 1a; 43,158 km$^2$) indicates that this dataset is representative of the range of CH$_4$ concentrations present in southern and mid-North Sea basal-peat deposits. The findings confirm the long-held hypothesis that methane CH$_4$ is stored in the southern North Sea basal-peat deposits formed during the Late Pleistocene and Early Holocene (Judd et al., 1997; Missiaen et al., 2002). These findings may indicate that basal-peats in other locations, particularly those from a similar period, have

the potential to function as CH$_4$ storage facilities and address an important gap in the inventory of global marine carbon and CH$_4$ budgets.

### 4.2 A newly measured carbon store





The study area spans 43,158 km$^2$, approximately ten percent of present-day European peatlands (Xu et al., 2018),
and larger than the surface area of the Netherlands. The total carbon stored in these submerged basal-peat
deposits is estimated to be 741 Tg C, corresponding to an average of 0.017 Tg C km$^{-2}$. Present-day Northern
peatlands have been estimated to store 547,000 Tg C, over a surface area of 4,000,000 km$^2$ (Yu et al., 2010). The
calculated amount of total carbon stored in North Sea basal-peat per km$^2$ is lower due to the decomposition of
carbon over time.


### 4.3 The CH$_4$ budget

Due to unattributable changes in atmospheric CH$_4$ concentrations in the last decade, quantification of the global
CH$_4$ budget has been a focal point of discussion in the literature, and wetland emissions are the largest source of
uncertainty (Saunois et al., 2020). In the present study, we estimated the volume of submerged basal-peat deposits
in the mid- and southern North Sea basin and the corresponding stored CH$_4$.

At all investigated sites, the methane concentrations in the basal-peat deposits were above the background
concentrations in bottom water as well as previously reported background concentrations of shallow sediments.
Due to a lack of published data, it is not possible to compare the CH$_4$ concentrations measured here with those of
other basal-peat deposits. Therefore, we compare the results of this study with the local water column and non-
peat sediments from the same basin as well as from basins in other parts of the world.

It is unclear whether the methane originates from within the peat deposits, and the contribution of methanogenic
microorganisms is difficult to quantify. Although the concentration of CH$_4$ harboured within the peat deposits was
low at most sites, the presence of CH$_4$ indicates that a local CH$_4$-producing mechanism is active. Previous research
indicates that deep methane seeps in the area are of biogenic origin. The activity of methanogenic microorganisms
accompanying the observed CH$_4$ concentrations suggests that methanogens are metabolizing some carbon into
CH$_4$ at a low rate. Sites with high CH$_4$ concentrations are likely indicative of deep methane seeps. This study
provides insights on methane stores at the landscape scale of basal-peat deposits in the North Sea and serves as
a step towards reducing uncertainties in the global CH$_4$ budget and better understanding the role of basal-peat
deposits in the global CH$_4$ cycle.

The CH$_4$ concentrations of 1-30 µmol l$^{-1}$ observed in the peat layer of the mid- and southern North Sea in the
present study are an order of magnitude higher than background concentrations measured in shallow North Sea
sediments (<0.1 µmol l$^{-1}$; (Niemann et al., 2005; Steinle et al., 2016) and much higher than concentrations observed
in the water column (maximum of 1.1 µmol l$^{-1}$ (Borges et al., 2016), with the exception of muddy sediments like
those of the Helgoland Bight, where observed CH$_4$ concentrations reached values of up to 6 mmol l$^{-1}$ (Aromokeye
et al., 2020). (Borges et al., 2016) reported average CH$_4$ concentrations in the water column of 0.139 µmol l$^{-1}$ (near-
shore) and 0.024 µmol l$^{-1}$ (off-shore) and a maximum concentration of 1.128 µmol l$^{-1}$. However, *in situ* CH$_4$
observations of peat sediments have not yet been measured because peat sediments are now inundated, buried
several meters beneath marine sediments.

The CH$_4$ concentrations measured in this study are higher than those measured in the water column in the same
area, due to the metabolism of CH$_4$ by methanogenic bacteria present in shallow sediments (Zhuang et al., 2018).
Darci's site is influenced by a known biogenic CH$_4$ gas seep located ±600 m beneath the seafloor (Schroot et al.,
2005). The CH$_4$ concentrations observed in these peat deposits are lower than but similar in magnitude to those





found in the near-surface sediments (less than 0.2 m beneath the sea floor) of a highly active gas seep in the northern North Sea (Niemann et al., 2005). The large methane concentrations observed at the Vittorio site may be the result of a deep gas seep.


### 4.4 Methanogenic but no methanotrophic communities

Oxic and anoxic batch incubations were used to assess both the $CH_4$ production and consumption potential of these basal-peat deposits. Methanogenesis was observed on methylated compounds only. In contrast to $H_2/CO_2$ and acetate, methylated compounds are a non-competitive methanogenic substrate that is metabolized by

*Methanosarcinales*, explaining the presence of these species in these sediments (Lyimo et al., 2000).

No aerobic or anaerobic methanotrophic prokaryotes were found in these peat deposits. Like many marine sediments, the anoxic and marine nature of the environment likely led to the exclusion of an aerobic methanotrophic population (Conrad et al., 1995). In addition, the low $CH_4$ partial pressure probably inhibited sulfate-dependent

anaerobic oxidation of $CH_4$ (Thauer, 2011). Sulfate reduction in these sediments is likely linked to $H_2$ and acetate oxidation (Oremland and Polcin, 1982). Environments with methanogens but not methanotrophs are uncommon but have occasionally been identified, e.g., in coal wells and masonry (Kussmaul et al., 1998; in 't Zandt et al., 2018). The absence of methanotroph activity is congruent with their absence in the results of 16S rRNA gene amplicon sequencing and confirms that methanotrophic species are most likely not present or active in this

environment.

### 4.5 Microbial communities of different submerged peat sites are diverse

We observed pronounced differences among the microbial populations at the four sampled locations (Vittorio, Max Gundelach, Senator Westphal S, and Westland sites). This heterogeneity indicates that the *in situ* microbial

populations in these deposits are influenced by the peat forming vegetation (Gastaldo et al., 2004; Stocker, 2012), in contrast to the homogenous results that would have been expected of an otherwise sedimentary-marine ecosystem. Although the sampling resolution was limited, this study provides the first insights into the microbial diversity of buried marine peat layers and confirms the relevance of the carbon source for the present-day microbial community composition (Fig. 6 & Fig. S4). Future studies with higher sampling resolution may provide a better

understanding of the relationship between plant and microbial species.

### 4.6 Plant succession is comparable in northern and southern regions of the study area

The patterns of vegetation succession at the Max Gundelach and Fredricksborg NE sites showed comparable local vegetation evolution, based on macro-analysis, and domination by mosses. The two plant successions of the Max

Gundelach and Fredricksborg NE sites do not disagree with the local plant succession documented by Wolters *et al*. (2010), located latitudinally (54°06.00', 06°46.70) between the Max Gundelach and Fredricksborg NE sites. The site of Wolters *et al*. (2010) was established with birch woodland, leading to carr vegetation during paludification, Salix and reeds during the establishment of a lagoonal environment, and finally Chenopodiaceae indicates the inundation of the marine environment. The results of this present study builds upon the findings of Wolters et al.

which did not assess changes in mosses, the primary peat-forming vegetation type.

The parallel sequences observed at the Max Gundelach and Fredricksborg NE sites begin and end at different times, suggesting that a comparable geomorphological context was present at both sites but during different periods. Moreover, the sequence of vegetation succession spanned c. 3000 years at the Max Gundelach site but





only 800 years at the Fredricksborg NE site, indicating that the changes in geomorphological conditions occurred at vastly differing rates. These periods are aligned with the peat growth described by Wolters et al. (2010) which prevailed for 1,300 years before inundation occurred. The differences in the duration and rate of peat accumulation are likely the result of the differences in the rate of sea-level rise between these two locations, in addition to other, largely unknown, palaeoenvironmental factors. The variation in $CH_4$ concentrations between sites is large and may

reflect the types of organic matter available to *in situ* microbial communities and subsequent mineralization rates.

### 4.7 Dominance of Bathyarchaeia suggests a central role in organic matter turnover

Bathyarchaeia dominated the archaeal communities of the peat sediments, with an average relative abundance of 70%. This phylum is an evolutionary diverse microbial group that is found in a wide range of organic-rich

environments, including deep sea and freshwater sediments (Evans et al., 2015). Bathyarchaeia often dominate marine subsurface archaeal communities, with relative abundances ranging from 10% to 100% (Fry et al., 2008; Zhou et al., 2018). Peat deposits are rich in cellulose and lignins (McMorrow et al., 2004), which are eventually converted to fluvic and humic acids that are more accessible to the microbial community (Bozkurt et al., 2001). The growth of Bathyarchaeota subgroup 8 (Bathy-8) on lignin suggests a key role of these species in the degradation

of peat organic material (Yu et al., 2018), and based on chemical rate estimation, they have been identified as one of the most active phyla in deep sea sediments (Fry et al., 2008). These findings support the high relative abundance observed in our study and the potential central role of Bathyarchaeia in the peat sediments. However, further experimental evidence is needed to confirm the role of Bathyarchaeia.

### 4.8 Lokiarchaea may play an important role in microbial fermentation


Lokiarchaeal sequences were highly abundant in the three samples of Vittorio Z, and this location also showed the highest $CH_4$ accumulation values (Fig. 6). Genome-based studies have indicated that their cellular machinery includes eukaryotic signature proteins, a cytoskeleton and phagocytic potential, suggesting Lokiarchaea are "missing link" microorganisms between prokaryotes and eukaryotes (Spang et al., 2015). Lokiarchaea have not

been previously detected in peat sediments, but a previous 16S rRNA gene and metagenome-based study of sub-seafloor sediments of the Costa Rica Margin also found Lokiarchaeota among the major microbial phyla; thus, Lokiarchaeota may be indicative of a marine environment (Martino et al., 2019). Genomic analyses of "Candidatus Lokiarchaeota" have indicated an acetogenic lifestyle, hydrogen dependency and mixotrophic potential (Sousa et al., 2016; Spang et al., 2019). Similarly, metabolic activity analyses of Namibian shelf sediments have revealed

potential for homoacetogenesis (Orsi et al., 2020). Populations of Lokiarchaea may provide important metabolic functions in organic matter degradation and methanogenic microbial guilds in marine sediments.

### 4.9 Candidate JS1 phylum bacteria dominate the potentially heterotrophic bacterial community

The JS1 lineage is a subgroup of the candidate phylum Atribacteria (Nobu et al., 2016). Metabolic reconstructions

indicate their potential for fermentative metabolism and syntrophic acetate oxidation (Lee et al., 2018), and several studies have indicated they are abundant within marine sediments (Fry et al., 2008; Lee et al., 2018). Studies in the Skagerrak, the German Wadden Sea and the Benguela Upwelling System showed that the upper sediment layers were mainly dominated by Delta- and Gammaproteobacteria, whereas deeper parts of the subseafloor were dominated by the JS1 lineage and Chloroflexi (Parkes et al., 2007; Wilms et al., 2006). This distribution is in line

with our findings of high relative abundances of JS1 lineage bacteria in the peat deposits (Fig. 6). A 16S rRNA PCR-DGGE study of two Wadden Sea tidal flats (Neuharlingersieler Nacken and Gröninger Plate) found that JS1 lineage bacteria were most abundant in the Neuharlingersieler Nacken samples with the highest total organic



carbon contents (1-2%) (Webster et al., 2007). Considering these previous findings of JS1 lineage bacteria in organic-rich environments, it is not unexpected that JS1 are dominant bacteria in these deep organic-rich peat
deposits.

### 4.10 Peat structure, CH₄ storage, and microbial community structure

The Vittorio site contained the maximum $CH_4$ concentrations observed in this study, both above and below the peat layer. Previous studies in non-peat marine environments have found $CH_4$ bubble emanations and dissolved $CH_4$
seepage vary strongly depending on sediment characteristics (Schneider von Deimling et al., 2015; Steinle et al., 2016). It is likely that the compacted nature of basal-peat deposits impacts the diffusion of $CH_4$ through the sediment (Grunwald et al., 2009).

Pore water analysis indicated conversion of the peat deposits into marine systems (Table S2); that is, marine
microbes have been introduced into sediments that previously harboured freshwater microbial communities. This is reflected by the occurrence of Dehalococcoidia and candidate phylum JS1 bacteria (labelled "JS1" in Fig. 6), which are characteristic of marine sediments. These species showed the highest abundances in the two sites located nearest to each other in the study area, Vittorio Z and Max Gundelach SW (Nobu et al., 2016; Wasmund et al., 2014). Candidate phylum JS1 bacteria were omnipresent, whereas Dehalococcoidia were more abundant in
three of the 12 layers, without a clear link to depth or local conditions. Future research is needed to unravel the mechanism underlying this pattern of abundance.

The high $CH_4$ concentrations observed at the Vittorio and Darci's sites occurred in the presence of nitrate and ammonium, previously shown to be indicative of increased rates of biological mineralization (Burdige, 1991).
However, the Westland site displayed comparably high ammonium concentrations without a link to $CH_4$ concentrations.

### 5.0 Conclusions

Methane concentrations were generally low with locally high concentrations. The source of methane remains unknown. Basal-peat deposits are globally widespread beneath land and ocean surfaces and function as a store of $CH_4$ that, in the event of physical disturbance, has been shown to be at risk of being released into the atmosphere. Large carbon stores in the presence of methanogenic bacteria in the absence of methanotrophs hold the potential for this material to be metabolised into methane gas, under different environmental conditions. The
results of this study are a stepping stone towards assessing the contribution of basal-peat deposits to regional and global carbon and methane budgets. Microbial community structure analysis using 16S rRNA gene-based sequencing techniques and activity assays indicated the absence of a $CH_4$ biofilter. This knowledge contributes to better understanding the roles and feedbacks of microorganisms in $CH_4$ storage and cycling.





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

**Data and materials availability:** All data needed to evaluate the conclusions in the paper are present in the paper and/or the appendices. Additional data related to this paper may be requested from the authors. Amplicon sequencing data were deposited in the GenBank database under the BioProject PRJNA639452.

**Author contributions:** Conceptualization: TL MitZ HD. Analysis: TL MitZ NvdP FB MH PvdV TdG ZvA. Funding acquisition: FB MH HN MJ CW HD. Methodology: TL MitZ NvdP OM CS HD. Writing – original draft: TL MitZ CW HD. Writing – review & editing: TL MitZ

**Competing interests:** The authors declare that they have no competing interests.

**Acknowledgements:** This work was supported by the Netherlands Organisation for Scientific Research through the
Soehngen Institute of Anaerobic Microbiology (SIAM) Gravitation Grant [grant number 024.002.002] and the Netherlands Earth System Science Center Gravitation Grant [grant number 024.002.001]. MSMJ was supported by ERC AG Ecomom 339880, and MSMJ and CPS were supported by ERC SyG Marix 854088. We thank Theo van Alen, Sihle Patience Ginindza, and Dave van Wes for technical assistance. We thank the Royal Netherlands Institute for Sea Research (NIOZ) and particularly Gert-Jan Reichhart, who was instrumental in organizing the two
cruises. We thank the captain, and the entire crew of the R/V Pelagia for enabling the success of the two sampling campaigns.