# Peer review of "Microbial activity, methane production, and carbon storage in Early Holocene North Sea peats"

_Biogeosciences, 2020_

## Referee Comment (RC1) · Rhiannon Mondav (Referee) · 28 Nov 2020

General comments: The authors have done a lot of work creating datasets to characterise the geochemistry, biogeochemistry, botany, microbial community, and carbon of submerged peats from the southern North Sea. Marine sediments are an understudied aspect of C cycle and climate and specifically the area of the North sea investigated here with its sandwiched layer of peat. The carbon pool of the region is estimated and potential emissions calculated. The potential for biological conversion of this carbon pool to CH4 is analysed via microbial taxonomic survey and incubations. This study shows a decoupling between CH4 production and C storage and posits that there is

threat of re-coupling.

In general, the ideas are clearly defined though a little more work connecting ideas within paragraphs in the introduction and connecting the theory presented in the introduction to what was done in the study will assist readers in following the work. Further, adding explanations and justification for method choices will also assist readers. It is currently unclear why certain data was collected and analyses done. Some of the datasets are underutilized. Analyses connecting the different datasets could increase the value of the manuscript (MS) to a variety of audiences. There are a few instances in the discussion that reference data or results not detailed in the results section.

Specific comments: Figure 1A states that the southern North sea was inundated due to anthropogenic caused sea level rise. Is this a typo? Figure 1C would be great if it was even higher resolution covering just the section sampled ie the rectangle in 1B, with sample locations marked. Maybe even the location of the 'special' samples coded in a different colour or symbol. Just to help the readers visualize what was done.

The methods section, especially the computational description needs more detail (or citations) and should include versions of software used and parameters chosen. -Why was sequencing and culture carried out on different samples/cores? It would make more sense to survey the community that was the base for incubations. Please provide justification for this choice. -I also ask for justification for choice of 60°C annealing temperature for the initial amplification. Please also check the citation for the bacterial reverse primer it is the same paper as for the forward. -I could not find the deposited sequence data as there was no listing in Genbank found for the BioProject identifier. -Why was a qPCR carried out? -2.6.3 why is cloning mentioned in the subtitle? Was cloning done? Please provide method, results, and justification for using this method. And integrate into results, discussion and conclusion. -The link to the core data also is not yet working.

I believe that the physical, chemical, botanical, and radiocarbon dating (others?) were

all done in order to establish how and when the peats formed and maybe what quality of carbon they hold. A large portion of the MS describes sampling, testing, physical qualities of the cores so it would be worth stating why these attributes were analysed as I did not notice this explicitly stated anywhere. Providing justification and motivation for choices will help the reader (who is unlikely to have the same level of expertise as the authorship team) to understand the work.

Published literature documents both co-occurrence and (spatial and temporal) separation of methanogenesis and methanotrophy. There is also substantial literature on the ANME archaea which I did not notice specifically and clearly mentioned in this MS. Here are some randomly chosen non-exhaustive examples for your consideration: https://sfamjournals.onlinelibrary.wiley.com/doi/10.1111/1462-2920.13096 https://www.ncbi.nlm.nih.gov/pmc/articles/PMC5104750/ https://pubmed.ncbi.nlm.nih.gov/30664670/ https://aem.asm.org/content/74/13/3985

I would love to see greater use made of the core chemical data. Perhaps a multivariate approach comparing microbial community membership to chemistry would be very interesting and tie together major components of the data presented here. Similarly, there a few places in the MS where the C, CH4, or organic content of the peats is referred to in comparison to microbes but I did not notice a statistical analysis to back up any comparison. This could add value to the MS.

I have concerns about the NMDSs presented. My reading of the manuscript is that there were 12 samples sequenced. On an NMDS where the samples are mapped onto species space there should therefore only be 12 dots. Please provide details of computational methods used so that what has been plotted on the NMDS is understood. For an example of the level of method detail required and correct plotting of an NMDS see e.g. https://www.pnas.org/content/115/47/11994

The MS states early on that it looks at C storage and CH4 seepage/accumulation etc. Please check the MS for typos 'CH4 storage' or do you have evidence that CH4 is

trapped in the peat? Is that why the term CH4 storage is used? I would have guessed that the CH4 currently in the peat deposits can bubble up through the overlying clay and sands into the water column where (depending on factors that might be worth listing) it is consumed by methanotrophs in the water column or emitted to the atmosphere. This could make an interesting discussion point for this MS.

There is a statement in the abstract and conclusion that the C in the peats could be converted to CH4 under other circumstances. What other circumstances? Your MS shows and states that the remaining C is not accessible to methanogens so what would make it available? This would be an interesting discussion point.

Technical corrections: -Community structure was not studied. Community membership was, please change this throughout MS. -There is a mix of 'methane' and 'CH4' throughout the MS please pick one. -Ln 779 methanogenic bacteria – is this a typo? -'activity assay' refers to e.g testing catalase activity in a lab. This study documents incubations not activity assays. Please be careful about using the word 'activity' (including in the title) throughout the MS.

---

## Referee Comment (RC2) · Anonymous Referee #2 · 19 Feb 2021

Note from reviewer: I do not have expertise in the experimental elements of this manuscript, thus my critique of microbial activity, gene sequencing and methane production is limited. General comments This manuscript documents a significant carbon store in the North Sea during the last glacial-interglacial transition, with experiments to understand the precise microbial activity and methane production/potential. The authors use gene-based sequencing to understand the microbial community structure and to explore the role and potential of peat microbial communities in carbon (methane) cycling. A key contribution of this work is quantifying these peats via observations/measurements and incubation experiments to determine the carbon storage potential with implications for better understanding the role of peat deposits in the

global carbon budget. This manuscript is well written and contributes important knowledge for better understating the role of buried peats in the carbon cycle. Two overall suggestions (1) keep discussion and result separate. For example, "capped by either shallow marine clay or sands" should perhaps move the 'shallow marine' interpretation to the discussion section. There are countless other examples of discussion embedded into the results section, for example starting at L477-L484. This would help with the organization of the manuscript. Also (2) perhaps place more emphasis on the global implications of this work for the carbon budget ie. How much does this change our estimates of carbon stores? What are the potentials for this carbon to be released in the future? Are there any other regions where a similar peat has been deposited? Specific comments L70 – "ice sheets reaching as far south as the Doggerbank area were subjected to strong glacio-isostatic adjustment" – ice sheets were subject to GIA? Or the earth was subject to GIA? please clarify. L95 – "task of measuring CH4 stores remains challenging" – why is this the case? L145 – why were these sites chosen for microbial sequencing? This is unclear. Do they provide good spatial coverage that is representative of the region? L213 – same as above. why were these particular sites chosen for microbial sequencing? This is unclear. L325-327 – this mixture of high/low sampling resolution and high/low taxonomic resolution is interesting. Perhaps an extra line on why this technique was chosen? L391 – " the thickness of the peat layer does not appear to play a determining role in CH4 concentrations, as both thick and thin peat layers harboured both high and low CH4 concentrations" – this is an interesting finding of this work, with implications for carbon modelling of paleo-peatlands. Is it possible to show this graphically? A quick plot showing thickness vs. CH4 concentration? L420 – The header suggests that this section will contain information on plant macrofossil communities, but there is no such information here. L489 – given the high spatial variability in peat thickness, I would expect to see (large?) errors on this estimate. What uncertainties were incorporated into this calculation and how do they impact the resulting error? L614 – what is meant by "from a similar period"? is this referring strictly to the time interval, or the sequence of events (SL rise) that would cause these peats to

be buried? Fig. 1 – Does the 3rd panel "North Sea basin" refer to present-day conditions? It might be worth clarifying. Fig. 1 caption – "The distribution of tsites within this sampling area" – sites?

---

## Author Comment (AC1) · 19 Mar 2021

Thank you for your comments and constructive suggestions. We provide a response to each comment below.

Reviewer 1: Rhiannon Mondav (Referee) rhiannon.mondav@ebc.uu.se

Reviewer 1 General comments: The authors have done a lot of work creating datasets to characterise the geochemistry, biogeochemistry, botany, microbial community, and carbon of submerged peats from the southern North Sea. Marine sediments are an

understudied aspect of C cycle and climate and specifically the area of the North sea investigated here with its sandwiched layer of peat. The carbon pool of the region is estimated and potential emissions calculated. The potential for biological conversion of this carbon pool to CH4 is analysed via microbial taxonomic survey and incubations. This study shows a decoupling between CH4 production and C storage and posits that there is threat of re-coupling.

In general, the ideas are clearly defined though a little more work connecting ideas within paragraphs in the introduction and connecting the theory presented in the introduction to what was done in the study will assist readers in following the work. Further, adding explanations and justification for method choices will also assist readers. It is currently unclear why certain data was collected and analyses done. Some of the datasets are underutilized. Analyses connecting the different datasets could increase the value of the manuscript (MS) to a variety of audiences. There are a few instances in the discussion that reference data or results not detailed in the results section.

R: We will include extra sentences to more clearly connect ideas in the introduction. We feel that throughout the paper, we use the theory to justify the study design, the methods, and to quantitatively compare and discuss our results in the discussion. We better explain our reasons why the macrofossils and DNA analysis was performed in this way. It is true that our datasets do not all receive an equal focus in this paper. Firstly, some data are more indicative of the ecosystem dynamics than others and secondly, some data is more novel than others. For this reason, this paper focuses on the microbiology analyses, CH4 analyses, and macrofossil analyses. We discuss the possibility of statistically comparing data in a later specific comment. We will remove L622-624 from the discussion because it does not add insightful knowledge, "Present-day Northern peatlands have been estimated to store 547,000 Tg C, over a surface area of 4,000,000 km2 (Yu et al., 2010). The calculated amount of total carbon stored in North Sea basal-peat per km2 is lower due to the decomposition of carbon over time."

Reviewer 1 Specific comments: Figure 1A states that the southern North sea was inundated due to anthropogenic caused sea level rise. Is this a typo?

R: We would like to change the 3rd panel of Fig 1 to "North Sea basin (present day conditions, human-induced climate warming, rising sea levels)". We will also change Fig 1 caption from "tsites" to "sites".

Reviewer 1: Figure 1C would be great if it was even higher resolution covering just the section sampled ie the rectangle in 1B, with sample locations marked. Maybe even the location of the 'special' samples coded in a different colour or symbol. Just to help the readers visualize what was done.

R: We've added a panel that shows the sample sites. Figure 1 caption: Peats submerged beneath the North Sea region of study. (A) Schematic of the evolution of processes that led to the conversion from the Pleistocene land surface to the buried marine peat sediments as they occur today. (B) The sampling area location within the context of Western Europe, (C) the sampling areas, and (D) the sampling sites in the North Sea, coloured according to the area names, plotted in C. B, C, and D were generated using Python's Basemap module and the background map image uses NASA's Earth Observatory's Blue Marble: Next Generation.

Reviewer 1: The methods section, especially the computational description needs more detail (or citations) and should include versions of software used and parameters chosen. -Why was sequencing and culture carried out on different samples/cores? It would make more sense to survey the community that was the base for incubations. Please provide justification for this choice. -I also ask for justification for choice of 60âŮęC annealing temperature for the initial amplification. Please also check the citation for the bacterial reverse primer it is the same paper as for the forward. -I could not find the deposited sequence data as there was no listing in Genbank found for the BioProject identifier. -Why was a qPCR carried out? -2.6.3 why is cloning mentioned in the subtitle? Was cloning done? Please provide method, results, and justification for

using this method. And integrate into results, discussion and conclusion. -The link to the core data also is not yet working.

R: -We have added detailed information on software versions and parameters in the materials and methods section. -Regarding the annealing temperature: the primers were tested in an annealing temperature gradient experiment before, and 60 degrees was determined as the optimal temperature. -There are indeed two papers for the two different primers: Herlemann et al. (2011) ISME J and Klindworth et al. (2013). Nucleic Acids Res -Indeed the sequence data is not yet available in Genbank, since the data deposit will be openly available upon publication. -The cloned 16S rRNA gene fragments were used as a standard in the qPCR. We have now included a respective citation where it is described how the respective plasmids were obtained. We have also included additional information on software analysis and qPCR efficiency. -Unfortunately, the cores from the first sampling expedition did not provide enough material to perform both sequencing and the incubation experiments. Therefore, we chose to divide the experiments over the different sites in order to obtain the maximum amount of information possible, while taking the experimental constraints into consideration. -We carried out a quantitative PCR to investigate the relative abundance of bacteria and archaea in these samples. This is especially relevant for microorganisms in the methane cycle, since all methanogenic microorganisms are found in the archaeal domain. Therefore, the qPCR results provide an indication of the relative contribution of methanogenic archaea in these ecosystems.

Reviewer 1: I believe that the physical, chemical, botanical, and radiocarbon dating (others?) were all done in order to establish how and when the peats formed and maybe what quality of carbon they hold. A large portion of the MS describes sampling, testing, physical qualities of the cores so it would be worth stating why these attributes were analysed as I did not notice this explicitly stated anywhere. Providing justification and motivation for choices will help the reader (who is unlikely to have the same level of expertise as the authorship team) to understand the work.

R: Thank you for your interest. We believe that this is articulated in the final paragraph of the introduction, "To provide a better understanding of the basal-peat ecosystem submerged beneath the North Sea, and its role in the CH4 cycle, we measure in situ CH4 concentrations and sediment organic matter content. Further, plant macrofossil analysis was performed to determine plant community composition and describe the micro-organismal habitat. 16S rRNA gene amplicon sequencing was performed to determine microbial diversity, and batch incubations were conducted to investigate actual and potential microbial CH4 cycle activity in the submerged peat deposits."

Reviewer 1: Published literature documents both co-occurrence and (spatial and temporal) separation of methanogenesis and methanotrophy. There is also substantial literature on the ANME archaea which I did not notice specifically and clearly mentioned in this MS. Here are some randomly chosen non-exhaustive examples for your consideration: https://sfamjournals.onlinelibrary.wiley.com/doi/10.1111/1462-2920.13096 https://www.ncbi.nlm.nih.gov/pmc/articles/PMC5104750/ https://pubmed.ncbi.nlm.nih.gov/30664670/ https://aem.asm.org/content/74/13/3985

R: While we agree with the referee that the topic of ANME activity and the interactions of ANME/aerobic methanotrophs and methanogens is a highly interesting research topic, we do not see the need to incorporate this into our manuscript. Both our incubation experiments as well as our amplicon study confirm that neither ANME methanotrophs nor aerobic bacterial methanotrophs were present in our samples. Therefore, adding additional literature on ANME/methanogen co-occurrence remains speculative. We have discussed the absence of methanotrophs in our discussion and conclusion.

Reviewer 1: I would love to see greater use made of the core chemical data. Perhaps a multivari- ate approach comparing microbial community membership to chemistry would be very interesting and tie together major components of the data presented here. Similarly, there a few places in the MS where the C, CH4, or organic content of the peats is referred to in comparison to microbes but I did not notice a statistical analysis to back up any comparison. This could add value to the MS.

R: We agree that a statistical comparison would substantiate discussion. We will include the following PCA figures for the archaea and bacterial populations.

L603 of results: "Principal Component Analysis (PCA) was used to describe the relationships between the large number of sampling methods. Each principal component (PC) is an uncorrelated linear combination of variables that maximises variance and the PC loads represent the relative contributions of the original variables to the PCs. PCA was calculated for archaeal (Fig. 6a) and bacterial (Fig. 6b) species abundance separately.

The arachael PC1 and PC2 account for 59% (PC1 accounts for 32.8% and PC2 accounted for 26.0%) of the total data variability of all variables included in this analysis. PC1 loadings showed that Marine Benthic Group is anti-correlated with OM content (labelled LOI330, LOI550). I.e. Marine Benthic Group B population abundance was greater in samples with lower OM content. PC2 loadings indicated Bathyarachaeia and Methanoregulaceae population abundance are anti-correlated with high methane concentrations and Lokiarchaeia population abundance.

The bacterial PC1 and PC2 accounted for 43% (PC1 accounts for 26.7% and PC2 accounted for 15.8%) of the total data variability of all variables included in this analysis. There were clear groupings indicated by both the PC1 and PC2 loadings. Methane concentration did not have strong (anti)correlation with any variable in any PC. JS1 was negatively correlated with Sprochaetales, SBR1031, Pla1 lineage, and Pirellulales in PC1 loadings."

Figure caption: Principal component analyses calculated using species' abundance, CH4 concentration ('CH4_porewater'), latitude, porosity,OM content (LOI330, LOI550) and depth beneath seafloor ('dbsf_m'). PC1 loadings (x-axis) are plotted against the PC2 loadings (y-axis) for A. archeal species' abundance, B. bacterial species' abundance.

Reviewer 1: I have concerns about the NMDSs presented. My reading of the

manuscript is that there were 12 samples sequenced. On an NMDS where the samples are mapped onto species space there should therefore only be 12 dots. Please provide details of computational methods used so that what has been plotted on the NMDS is under- stood. For an example of the level of method detail required and correct plotting of an NMDS see e.g. https://www.pnas.org/content/115/47/11994

R: We have corrected the legends of the NMDS plots to properly explain the procedure. The NMDS plots here are based on the OTUs that were pre filtered. OTUs that only occur once per sample (on average for the total amount of samples: OTUs with 12 or less occurrences were removed). The dots in these plots thus represent OTUs.

Reviewer 1: The MS states early on that it looks at C storage and CH4 seep-age/accumulation etc. Please check the MS for typos 'CH4 storage' or do you have evidence that CH4 is trapped in the peat? Is that why the term CH4 storage is used? I would have guessed that the CH4 currently in the peat deposits can bubble up through the overlying clay and sands into the water column where (depending on factors that might be worth listing) it is consumed by methanotrophs in the water column or emitted to the atmosphere. This could make an interesting discussion point for this MS.

R: CH4 ebullition is possible. Based on our observations, we do not understand it to be a widespread occurrence in this environment. We propose to include the following text into the discussion of the CH4 budget (L626 onwards): "There exists two potential hypotheses explaining the presence of CH4. Firstly, the CH4 observed here was produced some time ago and in the absence of active methanotrophs, has been trapped by the overlying sediment layer. The observed clay layer overlying the peat is sufficiently dense to prevent CH4 outgassing. Our results show that neither aerobic or anaerobic methanoptrophic prokaryotes were activated by oxic or anoxic incubations.

Alternatively, the observed CH4 concentrations were produced by methanogens in the present day. Our incubation activity studies show that whilst the methanogenic community could be revived within a two week window, methanogens were not observed

to be active in the present day (Fig. 5). Therefore, it is likely that trapped pockets of old methane have been observed here. This supports previous non-in situ seismic studies, showing trapped methane pockets in the sedimentary peat layer beneath the North Sea but contradicts the hypothesis that this methane is produced in the present day."

Reviewer 1: There is a statement in the abstract and conclusion that the C in the peats could be converted to CH4 under other circumstances. What other circumstances? Your MS shows and states that the remaining C is not accessible to methanogens so what would make it available? This would be an interesting discussion point.

R: "Methanotrophs have the potential to be activated in the presence of additional CH4. Such additional CH4 may occur due to emission caused by leakage from fossil fuel extraction, which has occurred in the local area previously (Schneider von Deimling et al. 2015). Upon activation, methanotrophs would have the potential to consume both the newly added and existing methane sources."

Reviewer 1: Technical corrections: -Community structure was not studied. Community member- ship was, please change this throughout MS.

R: The term 'community structure' is commonly used in the literature to describe the results of amplicon-based sequencing studies. For this reason, we also use this term in our manuscript.

Reviewer 1: -There is a mix of 'methane' and 'CH4' throughout the MS please pick one.

R: Thank you for picking this up. L612: "The findings confirm the long-held hypothesis that methane CH4 is stored...." is changed to, "The findings confirm the long-held hypothesis that CH4 is stored..." We will change 'methane' to 'CH4' except for where it occurs in a heading or at the start of a sentence.

Reviewer 1: -Ln 779 methanogenic bacteria – is this a typo?

R: Thank you for spotting this. "Large carbon stores in the presence of methanogens

but in the absence of methanotrophs hold the potential to be metabolised into methane gas..." We will correct any other instances.

Reviewer 1: -'activity assay' refers to e.g testing catalase activity in a lab. This study documents incubations not activity assays. Please be careful about using the word 'activity' (in- cluding in the title) throughout the MS

R: To avoid potential confusion we have changed the occurrences of 'activity assays' with 'incubations'
* * *
[Figure]

**Fig. 1.**

[Figure]

**Fig. 2.**

[Figure]

**Fig. 3.**

---

## Author Comment (AC2) · 19 Mar 2021

We thank the reviewer #2 for their considered and constructive comments on this manuscript. We provide a response to each comment below.

Reviewer 2: Anonymous Referee #2

Reviewer 2 Note from reviewer: I do not have expertise in the experimental elements of this manuscript, thus my critique of microbial activity, gene sequencing and methane pro- duction is limited. General comments This manuscript documents a significant carbon store in the North Sea during the last glacial-interglacial transition, with experiments to understand the precise microbial activity and methane production/potential. The authors use gene-based sequencing to understand the microbial community structure and to explore the role and potential of peat microbial communities in carbon (methane) cycling. A key contribution of this work is quantifying these peats via observations/measurements and incubation experiments to determine the carbon storage potential with implications for better understanding the role of peat deposits in the global carbon budget. This manuscript is well written and contributes important knowledge for better understating the role of buried peats in the carbon cycle. Two overall suggestions (1) keep discussion and result separate. For example, "capped by either shallow marine clay or sands" should perhaps move the 'shallow marine' interpretation to the discussion section. There are countless other examples of discussion embedded into the results section, for example starting at L477-L484. This would help with the organization of the manuscript. Also (2) perhaps place more emphasis on the global implications of this work for the carbon budget ie. How much does this change our estimates of carbon stores? What are the potentials for this carbon to be released in the future? Are there any other regions where a similar peat has been deposited?

R: When writing the manuscript, we found that limited interpretation of the results was required when explaining analyses that compare different experimental techniques (e.g. Fig. S4). We have revised the results section to keep this to a minimum.

L358: In the geological literature, it is common practice that lithographic descriptions include the sediment origin. Therefore, it is our opinion that "capped by either shallow marine clay or sands" is an observation, not an interpretation. We will modify L358 to reflect that this description is observational: "The localized nature of this landscape is apparent in the lithographic differences observed between and within sites (Fig 2). Peat deposits lie upon Pleistocene sands, capped by marine clays, at all sites. At some sites the overlying clay layer is stratified by marine sands (i.e. Dorthea Shallow SW, Dorthea SSW, Dorthea NW, Fredricksborg NE, Fig. 2) ."

We will move L170-L174, " Peat was recovered at all sites, except Easting Down, Stormvogel, and Darci's site." to L357 of the results section.

We will move L361-364, "The basal-peat developed due to rising groundwater as a result of the postglacial sea-level rise and was quickly capped by rapidly deposited clays and subsequent sand deposits in most instances or directly capped by sand in others." to create a new paragraph at L605 the discussion. We would like to add the following sentence to this paragraph, "We hypothesise that rapid flooding of the peatland led to the generation of large volumes of methane. In instances where this methane has neither escaped (by ebullition) or been consumed (by methanogens), it remains trapped by the porosity of the peat. Future studies may consider isotopic analysis as an indicator of origin."

Move: L390-L393, "The highest concentration of $CH_4$ was observed at the Vittorio site, at the latitude of Vlieland. The Vittorio site had the second thickest peat layer in this study, but the thickness of the peat layer does not appear to play a determining role in $CH_4$ concentrations, as both thick and thin peat layers harboured both high and low $CH_4$ concentrations." to L631 of the discussion.

Move L409-L12, "The period of active peat formation depended on the ability of peat formation to keep up with the rising groundwater table and on hydrological conditions; e.g., peat formation commenced earlier in areas with a less permeable substrate than in areas with a sandy substrate" to L701 of the discussion.

Move L477-483, "It is striking that the same three-step bryophyte dominated sequence of Sphagnum-Tomentypnum nitens-Warnstorfia/Drepanocladus occurs in both geo-graphically as well as temporally different sites. However, in contrast to the sequence of the Max Gundelach site, where Sphagnum magellanicum is present only at the start of the sequence, Sphagnum papillosum is present throughout the peat deposits at the Fredricksborg NE site. In general, plant remains are better preserved in the layers dominated by Sphagnum spp. than in those dominated by brown mosses." to L691 the discussion.

To quantify this carbon budget in relation to other carbon stock, we will add the following to L625, "The 741 Tg C stored in these submerged peats is equivalent to 70% of the C stored in Dutch peatlands today (1,030 Tg-C), and equivalent to 2.4% of the C stored in globally, the largest peatland C storage facility, the Congo Basin complex (30,600 Tg-C, Dargie et al., 2017)." We will add the following sentence to explain the mechanisms that could release this carbon into the atmosphere, "This carbon store has the potential to be released into the overlying water column in the occurrence of a marine seep, that could be either naturally initiated or an outcome of fossil fuel extraction (see Schneider von Deimling et al. 2015)."

We will quantify the CH4 budget in relation to global CH4. We will add the following to L632, "If released in one go, the 0.411 Tg-CH4, stored in these submerge peats is equivalent to approximately one quarter of the annual biogenic oceanic CH4 emissions (2 Tg yr-1), and one month of the CH4 emissions from all oceanic sources that were reported in 2000-2009. If released in one go, this amount is equivalent to approximately one month of the global atmospheric CH4 growth reported for the years, 2000-2009 (5.8Tg yr-1) or 1.5 weeks of the global atmospheric CH4 growth that occurred in 2017 (16.8 Tg yr-1)."

Reviewer 2 Specific comments Reviewer 2 L70 – "ice sheets reaching as far south as the Doggerbank area were subjected to strong glacio-isostatic adjustment" – ice sheets were subject to GIA? Or the earth was subject to GIA? please clarify.

R: Thank you for picking this up. We will edit the sentence to, "During the Last Glacial Maximum, the basin floor of the study area was subjected to strong glacio-isostatic adjustments (Vink et al., 2007)."

Reviewer 2 L95 – "task of measuring CH4 stores remains challenging" – why is this the case?

R: We will add the following sentence to L95. "Despite extensive efforts to map basal-peats at the global scale in recent decades (Treat et al., 2019; Xu et al., 95 2018), most

basal-peats are submerged beneath ocean sediments which are hard to reach, meaning accessing and measuring CH4 stores remains challenging (Dean et al., 2018)."

Reviewer 2 L145 – why were these sites chosen for microbial sequencing? This is unclear. Do they provide good spatial coverage that is representative of the region?

R: We chose to divide the experiments over the different sites in order to obtain the maximum amount of information possible, while taking the experimental constraints into consideration. Due to limitations in the available sample amounts we were not able to carry out both the incubation studies and the amplicon sequencing on the cores of the first sampling expedition. We will clarify in L394-396: "Unfortunately, the cores from the first sampling expedition did not provide enough material to perform both sequencing and the incubation experiments. Therefore, four sites (Fig. 2a-d) representing both high and low CH4 concentrations were chosen from the 2017 cruise (southern North Sea) for 16S rRNA gene-based sequencing to unravel the microbial community structure. Subsequent to the 2018 cruise (mid-Northern North Sea), a second cluster of four sites (Fig. 2e-f) were chosen to investigate the role and potential of in situ microbial communities in CH4 cycling."

Reviewer 2 L213 – same as above. why were these particular sites chosen for microbial sequencing? This is unclear.

R: As above.

Reviewer 2 L325-327 – this mixture of high/low sampling resolution and high/low taxonomic resolution is interesting. Perhaps an extra line on why this technique was chosen?

R: We propose to replace L325-L327 with the following text: "The Max Gundelach site, was analysed with low sample resolution but high taxonomic resolution, showing the main peat components as well as an overview of the less abundant taxa. As the less abundant taxa were, in this research, not highly relevant we analysed the Fredericksborg NE site with high sample resolution but low taxonomic resolution, showing only the main peat components. The sites can be compared based on the main peat components."

Reviewer 2 L391 – " the thickness of the peat layer does not appear to play a determining role in CH4 concentrations, as both thick and thin peat layers harboured both high and low CH4 concentrations" – this is an interesting finding of this work, with implications for carbon modelling of paleo-peatlands. Is it possible to show this graphically? A quick plot showing thickness vs. CH4 concentration?

R: We will adjust Figure 3, the methane depth profiles, to include the peat thickness and will adjust the figure caption as follows: "Depth profiles of methane concentrations at all sites in $\mu$mol L-1. The yellow line indicates the average methane concentration across all measurements. The green line indicates the average methane concentrations of seawater, measured in the same area (Borges et al., 2016). The pink shaded regions are indicative of peat. Note the varying y axes."

Reviewer 2 L420 – The header suggests that this section will contain information on plant macrofossil communities, but there is no such information here.

R: Section 3.4 includes the following subsections, consistent with the Biogeosciences guidelines: '3.4.1 Local vegetation succession in the southern North Sea', '3.4.2 Local vegetation succession in the mid North Sea at Doggerbank', '3.4.3 Local vegetation succession is analogous across sites'.

Reviewer 2 L489 – given the high spatial vari- ability in peat thickness, I would expect to see (large?) errors on this estimate. What uncertainties were incorporated into this calculation and how do they impact the result- ing error?

R: We will include the following after L185 of the methods, "The minimum and maximum peat layer thickness' were used to calculate the lower, and upper estimates of total peat volume, and total CH4 minimum and maximum error using the following formula: Total

CH4 = V * CH4 Where V is peat volume (m-3), CH4 concentration ($\mu$mol L-1) was calculated using measured sediment porosity, and the mean CH4 concentration was calculated using all samples."

Reviewer 2 L614 – what is meant by "from a similar period"? is this referring strictly to the time interval, or the sequence of events (SL rise) that would cause these peats to be buried?

R: We will clarify this sentence: "These findings may indicate that other basal-peats that formed during the LGM, have the potential to function as CH4 storage facilities and address an important gap in the inventory of global marine carbon, CO2, and CH4 budgets."

Fig. 1 – Does the 3rd panel "North Sea basin" refer to present-day condi- tions? It might be worth clarifying. Fig. 1 caption – "The distribution of tsites within this sampling area" – sites?

R: We would like to change the 3rd panel of Fig 1 to "North Sea basin (present day conditions, human-induced climate warming, rising sea levels)". Fig 1 caption should indeed be, "sites" and not "tsites". Thank you for picking this up.

We have used a new citation that we will include in the reference list: Dargie, G., Lewis, S., Lawson, I. et al. Age, extent and carbon storage of the central Congo Basin peatland complex. Nature 542, 86–90 (2017). https://doi.org/10.1038/nature21048

———————————————————

[Figure]

**Fig. 1.**

---

## Author Response (AR1)

**Response to Reviewer 1**

Thank you for your comments and constructive suggestions. We feel these comments were very helpful in improving the manuscript. We provide a response (in blue text) to each comment below.

**Reviewer 1:**

Rhiannon Mondav (Referee) rhiannon.mondav@ebc.uu.se Received and published: 28 November 2020

Reviewer 1 General comments: The authors have done a lot of work creating datasets to characterise the geochemistry, biogeochemistry, botany, microbial community, and carbon of submerged peats from the southern North Sea. Marine sediments are an understudied aspect of C cycle and climate and specifically the area of the North sea investigated here with its sandwiched layer of peat. The carbon pool of the region is estimated and potential emissions calculated. The potential for biological conversion of this carbon pool to CH4 is analysed via microbial taxonomic survey and incubations. This study shows a decoupling between CH4 production and C storage and posits that there is threat of re-coupling.

In general, the ideas are clearly defined though a little more work connecting ideas within paragraphs in the introduction and connecting the theory presented in the intro- duction to what was done in the study will assist readers in following the work. Further, adding explanations and justification for method choices will also assist readers. It is currently unclear why certain data was collected and analyses done. Some of the datasets are underutilized. Analyses connecting the different datasets could increase the value of the manuscript (MS) to a variety of audiences. There are a few instances in the discussion that reference data or results not detailed in the results section.

R: We have made some edits to the introduction to more clearly connect ideas.

We considered your point regarding justification of the study. We include a paragraph at L123 that clearly links the theory and our study aims and design.

We better explain our reasons why the macrofossils and DNA analysis was performed in this way at L215 of the methods.

It is true that not all datasets receive an equal focus in this paper. Firstly, some data are more indicative of the ecosystem dynamics than others and secondly, some data is more novel than others. For this reason, this paper focuses on the microbiology analyses, CH4 analyses, and macrofossil analyses.

We discuss the possibility of statistically comparing data in a reply to a specific comment.

We have removed L622-624 from the discussion because it does not contribute insightful knowledge.

**Reviewer 1 Specific comments:**

Figure 1A states that the southern North sea was inundated due to anthropogenic caused sea level rise. Is this a typo?

*R:* To clarify this, we have changed the heading to read, 'The submersion of north sea peatlands and the current microbial habitats'. We have changed the caption of the 3rd panel to, "North Sea basin present day conditions (human-induced climate warming, rising sea levels)".

Reviewer 1: Figure 1C would be great if it was even higher resolution covering just the section sampled ie the rectangle in 1B, with sample locations marked. Maybe even the location of the 'special' samples coded in a different colour or symbol. Just to help the readers visualize what was done. *R: We've added a panel that shows the sample sites and adjusted the figure caption, accordingly.*

Reviewer 1: The methods section, especially the computational description needs more detail (or citations) and should include versions of software used and parameters chosen. -Why was sequencing and culture carried out on different samples/cores? It would make more sense to survey the community that was the base for incubations. Please provide justification for this choice. -I also ask for justification for choice of 60°C annealing temperature for the initial amplification. Please also check the citation for the bacterial reverse primer it is the same paper as for the forward. -I could not find the deposited sequence data as there was no listing in Genbank found for the BioProject identifier. -Why was a qPCR carried out? -2.6.3 why is cloning mentioned in the subtitle? Was cloning done? Please provide method, results, and justification for using this method. And integrate into results, discussion and conclusion. -The link to the core data also is not yet working.

-We have added detailed information on software versions and parameters in section 2.6 of the materials and methods section.

*-Regarding the annealing temperature: the primers were tested in an annealing temperature gradient experiment before, and 60 degrees was determined as the optimal temperature.*

-There are indeed two papers for the two different primers: Herlemann et al. (2011) ISME J and Klindworth et al. (2013). Nucleic Acids Res

-Indeed the sequence data is not yet available in Genbank, since the data deposit will be openly available upon publication.

-The cloned 16S rRNA gene fragments were used as a standard in the qPCR. We have now included a respective citation where it is described how the respective plasmids were obtained. We have also included additional information on software analysis and qPCR efficiency.

-Unfortunately, the cores from the first sampling expedition did not provide enough material to perform both sequencing and the incubation experiments. Therefore, we chose to divide the experiments over the different sites in order to obtain the maximum amount of information possible, while taking the experimental constraints into consideration.

-We carried out a quantitative PCR to investigate the relative abundance of bacteria and archaea in these samples. This is especially relevant for microorganisms in the methane cycle, since all methanogenic microorganisms are found in the archaeal domain. Therefore, the qPCR results provide an indication of the relative contribution of methanogenic archaea in these ecosystems.

Reviewer 1: I believe that the physical, chemical, botanical, and radiocarbon dating (others?) were all done in order to establish how and when the peats formed and maybe what quality of carbon they

hold. A large portion of the MS describes sampling, testing, physical qualities of the cores so it would be worth stating why these attributes were analysed as I did not notice this explicitly stated anywhere. Providing justification and motivation for choices will help the reader (who is unlikely to have the same level of expertise as the authorship team) to understand the work.

*R*: Thank you for your interest. We explicitly state this in the final paragraph of the introduction (L123-L133) and in the first paragraph of the methods (L135-141).

Reviewer 1: Published literature documents both co-occurrence and (spatial and temporal) separation of methanogenesis and methanotrophy. There is also substantial literature on the ANME archaea which I did not notice specifically and clearly men- tioned in this MS. Here are some randomly chosen non-exhaustive examples for your consideration:

https://sfamjournals.onlinelibrary.wiley.com/doi/10.1111/1462-2920.13096

https://www.ncbi.nlm.nih.gov/pmc/articles/PMC5104750/ https://pubmed.ncbi.nlm.nih.gov/30664670/ https://aem.asm.org/content/74/13/3985

*R*: While we agree with the referee that the topic of ANME activity and the interactions of ANME/aerobic methanotrophs and methanogens is a highly interesting research topic, we do not see the need to incorporate this into our manuscript. Both our incubation experiments as well as our amplicon study confirm that neither ANME methanotrophs nor aerobic bacterial methanotrophs were present in our samples. Therefore, adding additional literature on ANME/methanogen co-occurrence remains speculative. We have discussed the absence of methanotrophs in our discussion and conclusion.

Reviewer 1: I would love to see greater use made of the core chemical data. Perhaps a multivari- ate approach comparing microbial community membership to chemistry would be very interesting and tie together major components of the data presented here. Similarly, there a few places in the MS where the C, CH4, or organic content of the peats is referred to in comparison to microbes but I did not notice a statistical analysis to back up any comparison. This could add value to the MS.

*R:* We agree that a multi-variable statistical comparison is useful and considered this point very seriously. As a result, we did perform a PCA analysis. In the end, we have decided not to include this in the final manuscript. We include the results here for your interest. We have removed and reduced comparing relationships between variables. Considering the highly heterogenous nature of peat OM, and micro-organisms, our sample size is not adequate to deduce relationships or processes.

*Figure description*: Principal component analyses calculated using species' abundance, CH4 concentration, porosity, OM content (LOI330, LOI550) and depth beneath seafloor (dbsf), and site. PC1 loadings (x-axis) are plotted against the PC2 loadings (y-axis) for **A**. archeal species' abundance, **B**. bacterial species' abundance. The coloured dots represent microbial species and uses the legend of Fig. 5b.

Reviewer 1: I have concerns about the NMDSs presented. My reading of the manuscript is that there were 12 samples sequenced. On an NMDS where the samples are mapped onto species space there should therefore only be 12 dots. Please provide details of computational methods used so that what

has been plotted on the NMDS is under- stood. For an example of the level of method detail required and correct plotting of an NMDS see e.g. https://www.pnas.org/content/115/47/11994

*R*: We have corrected the legends of the NMDS plots to properly explain the procedure. The NMDS plots here are based on the OTUs that were pre filtered. OTUs that only occur once per sample (on average for the total amount of samples: OTUs with 12 or less occurrences were removed). The dots in these plots thus represent OTUs.

Reviewer 1: The MS states early on that it looks at C storage and CH4 seepage/accumulation etc. Please check the MS for typos 'CH4 storage' or do you have evidence that CH4 is trapped in the peat? Is that why the term CH4 storage is used? I would have guessed that the CH4 currently in the peat deposits can bubble up through the overlying clay and sands into the water column where (depending on factors that might be worth listing) it is consumed by methanotrophs in the water column or emitted to the atmosphere. This could make an interesting discussion point for this MS. *R: We have discussed this in section '4.1 Origins of this newly measured CH4 store'.*

Reviewer 1: There is a statement in the abstract and conclusion that the C in the peats could be converted to CH4 under other circumstances. What other circumstances? Your MS shows and states that the remaining C is not accessible to methanogens so what would make it available? This would be an interesting discussion point.

*R:* We have added the following to L681 in the discussion: "Methanotrophs have the potential to be activated in the presence of additional CH4. Such additional CH4 may occur due to emission caused by leakage from fossil fuel extraction, which has occurred in the local area previously (Schneider von Deimling et al., 2015). Upon activation, methanotrophs would have the potential to consume both the newly added and existing CH4 sources."

Reviewer 1: Technical corrections: -Community structure was not studied. Community member- ship was, please change this throughout MS.

*R*: The term 'community structure' is commonly used in the literature to describe the results of amplicon-based sequencing studies. For this reason, we also use this term in our manuscript.

Reviewer 1: -There is a mix of 'methane' and 'CH4' throughout the MS please pick one. *R: Thank you for picking this up. L612: "… the long-held hypothesis that methane CH4 is stored…." is changed to, "… the long-held hypothesis that CH4 is stored…" We have changed 'methane' to 'CH4' except for where it occurs in a heading or at the start of a sentence.*

Reviewer 1: -Ln 779 methanogenic bacteria – is this a typo?

*R*: Thank you for spotting this. We have changed this to: "Large carbon stores in the presence of methanogens but in the absence of methanotrophs hold the potential to be metabolised into methane gas..." We have corrected any other instances.

Reviewer 1: -'activity assay' refers to e.g testing catalase activity in a lab. This study documents incubations not activity assays. Please be careful about using the word 'activity' (in- cluding in the title) throughout the MS

*R*: To avoid potential confusion we have changed the occurrences of 'activity assays' with 'incubations'

**Response to Reviewer 2**

We thank the reviewer #2 for their considered and constructive comments on this manuscript. They have been very helpful in improving the manuscript. We provide a response (in blue text) to each comment below.

**Reviewer 2:**

Anonymous Referee #2 Received and published: 19 February 2021

Reviewer 2 Note from reviewer: I do not have expertise in the experimental elements of this manuscript, thus my critique of microbial activity, gene sequencing and methane pro- duction is limited. General comments This manuscript documents a significant carbon store in the North Sea during the last glacial-interglacial transition, with experiments to understand the precise microbial activity and methane production/potential. The authors use gene-based sequencing to understand the microbial community struc- ture and to explore the role and potential of peat microbial communities in carbon (methane) cycling. A key contribution of this work is guantifying these peats via observations/measurements and incubation experiments to determine the carbon stor- age potential with implications for better understanding the role of peat deposits in the global carbon budget. This manuscript is well written and contributes important knowl- edge for better understating the role of buried peats in the carbon cycle. Two overall suggestions (1) keep discussion and result separate. For example, "capped by either shallow marine clay or sands" should perhaps move the 'shallow marine' interpretation to the discussion section. There are countless other examples of discussion embedded into the results section, for example starting at L477-L484. This would help with the organization of the manuscript. Also (2) perhaps place more emphasis on the global implications of this work for the carbon budget ie. How much does this change our estimates of carbon stores? What are the potentials for this carbon to be released in the future? Are there any other regions where a similar peat has been deposited?

**R:**

(1) We have revised the results section to keep this to a minimum.

L358: We have edited the paragraph at L358 to reflect that this description is observational. It is now at L365. We also changed the subheading to '3.1 Basal peats vary in thickness, formed on *Pleistocene sands, capped by marine clays.*'

We have moved L361-364, "The basal-peat developed due to rising groundwater as a result of the postglacial sea-level rise and was capped by rapidly deposited clays and in some instances, stratified sand deposits." to section, '4.1 Origins of this newly measured CH4 store' in the discussion.

Moved L390-L393 to L653 of the discussion and reword to: The highest CH4 concentrations were measured at the Vittorio site, the site of the second thickest peat layer. However, we did not find evidence that the thickness of the basal peat was linked to CH4 concentrations, as both thick and thin peat layers harboured both high and low concentrations (Fig. 3)."

Moved L477-483 to section 4.6 the discussion.

We removed L409-L412 because this is repeated in section 4.6 of the discussion.

In a similar spirit, we moved L170-L174 (methods), "Peat was recovered at all sites, except Easting Down, Stormvogel, and Darci's site." to L369 of the results section.

(2) We have placed our C and CH4 estimates in relation to other global accounting estimates. We have added the following to L657, "The 741 Tg-C stored in these submerged peats is equivalent to 70% of the 1,030 Tg-C stored in Dutch peatlands today (Erkens et al., 2016), or 2.4% of the 30,600 Tg-C stored in the globe's largest peatland C storage facility, the Congo Basin complex (Dargie et al., 2017). This C has the potential to be released into the overlying water column in the occurrence of a marine seep, that could be either naturally initiated or an outcome of fossil fuel extraction (Schneider von Deimling et al., 2015)." We have quantified the CH4 budget in relation to global CH4. We have added the following to L628, "For comparison with global CH4 inventories, the estimated 0.411 Tg CH4 present in these submerge sediments is equivalent to almost one quarter of the annual biogenic oceanic CH4 emissions (2 Tg-CH4 yr1) (Saunois et al., 2020a), 1 month of the global growth of atmospheric CH4 that occurred during the years, 2000-2009 (5.8Tg yr1), or 1.5 weeks of the global atmospheric CH4 growth that occurred in 2017 (16.8 Tg yr1) (Saunois et al., 2020a)."

**Reviewer 2 Specific comments**

Reviewer 2 L70 – "ice sheets reaching as far south as the Doggerbank area were subjected to strong glacio-isostatic adjustment" – ice sheets were subject to GIA? Or the earth was subject to GIA? please clarify.

TL: Thank you for picking this up. We have edited the sentence to, "During the Late Pleistocene and Early Holocene, strong glacio-isostatic adjustments (GIA) resulted in isostatic subsidence of the North Sea basin (Hijma et al., 2012; Vink et al., 2007) and, combined with rapid melting of polar ice sheets, high rates of sea level rise, up to 1-2 cm yr-1 (Hijma and Cohen, 2019), gave rise to paludification, peatland development and later peatland submersion (now basal peats)."

Reviewer 2 L95 – "task of measuring CH4 stores remains challenging" – why is this the case? *TL:* We have changed L95 to: "Despite extensive efforts to map these submerged peatland ecosystems (Treat et al., 2019; Xu et al., 2018), basal peats remain hard to reach, meaning accessing and measuring CH4 stores remains challenging, limiting in situ measurements, (Dean et al., 2018)."

**Reviewer 2 L145 – why were these sites chosen for microbial sequencing? This is unclear. Do they provide good spatial coverage that is representative of the region?**

*R:* We chose to divide the experiments over the different sites in order to obtain the maximum amount of information possible, while taking the experimental constraints into consideration. Due to limitations in the available sample amounts we were not able to carry out both the incubation studies and the amplicon sequencing on the cores of the first sampling expedition. We clarified this in L216-L220: "Four cores in the southern North Sea were selected for 16S rRNA amplicon sequencing, and 4 cores from the Doggerbank area were selected for microbial activity studies. Unfortunately, the cores from the first sampling expedition experiments. Therefore, we chose to divide the microbial experiments over multiple sites in order to obtain the maximum amount of information possible, whilst taking the experimental constraints into consideration."

Reviewer 2 L213 – same as above. why were these particular sites chosen for microbial sequencing? This is unclear.

R: As above.

Reviewer 2 L325-327 – this mixture of high/low sampling resolution and high/low taxonomic resolution is interesting. Perhaps an extra line on why this technique was chosen?

*R:* We have explained this in L332-337: "The Max Gundelach site was analysed with low sample resolution but high taxonomic resolution, showing the main peat components as well as an overview of the less abundant taxa. As the less abundant taxa were, in this research, not highly relevant we

analysed the Fredericksborg NE site with high sample resolution but low taxonomic resolution, showing only the main peat components. The sites can be compared based on the main peat components."

Reviewer 2 L391 – " the thickness of the peat layer does not appear to play a determining role in CH4 concentrations, as both thick and thin peat layers harboured both high and low CH4 concentrations" – this is an interesting finding of this work, with implications for carbon modelling of paleo-peatlands. Is it possible to show this graphically? A quick plot showing thickness vs. CH4 concentration? *R: We have adjusted Figure 3, the methane depth profiles, to include the peat thickness.*

---

## Referee Report (RR1)

The authors have done a lot of excellent work both responding to reviewers comments that will assist readers to understand their work. It is nicely rounded and very well placed into context.

I have a few points for consideration on the manuscript.

1. The authors have an incorrect reference for the Bacterial reverse primer. Please see the following quotes from the Klindworth paper cited for the reverse primer: "Per sample, two separate PCR reactions were performed in order to test two bacterial primer pairs for 16S rDNA amplification. Primer pairs were: (i): S-D-Bact-0341-b-S-17, 5′-CCTACGGGNGGCWGCAG-3′ (32), and S-D-Bact-0785-a-A-21, 5′-GACTACHVGGGTATCTAATCC-3 (32);"

"32
Herlemann D.P.R., Labrenz M., Juergens K., Bertilsson S., Waniek J.J., Anderrson A.F.. Transition in bacterial communities along the 2000 km salinity gradient of the Baltic Sea, *ISME J*, 2011, vol. 5 (pg. 1571-1579)"

2. In previous work by the authors in which these primers were tested they used a temperature gradient PCR to select 60° C. It is not stated what the authors observed at 60°C that helped them choose this temperature. Could the authors please include this information in the methods as it is relevant to the interpretation of the results and might help others in their lab work.

3. Generally an annealing temperature of +/- 5° from the mean Tm of the primer pair, and as low as possible to achieve amplification is chosen. Typically this particular primer pair is run with annealing temperatures between 48°C and 56°C. The use of a higher annealing temperature could increase amplification bias reducing the phylogenetic range of template DNA amplified. Please carefully phrase any statements about the absence of detection of any clade as the absence could, more than usual, be due to amplification bias.
-https://academic.oup.com/femsec/article/60/2/341/584515
-https://academic.oup.com/nar/article/41/1/e1/1164457#119410076
-http://cshprotocols.cshlp.org/content/2009/4/pdb.ip66.short?
casa_token=h2nXrvzrHDIAAAAA:4IzeOXuwzgz48-
voKEUGXXMXq7jPlHPUUAkmR8WSgcMNN3B79NXvMPf_nkNgxkTC7SbgD5TJQ5I

4. Please provide standard details of the PCR e.g. volumes, concentrations etc of reactants and template, or provide a reference where these details are given.

5. I have trouble reconciling the NMDSs in fig S7 which show a random placement of OTUs and figure 6 which shows that there is not just a difference in archaea and bacterial assemblages between sites but that differences can be seen at family and higher taxonomic levels and so should be very obvious at OTU level. Please provide details of how the NMDS were plotted as this difference seems strange. The

statement that no difference in structure was observed according to the NMDSs while an accurate description of what is plotted (though because sites are not plotted on the NMDSs the statement is only partially supported), does not fit with the other analyses which show clear differences at sites in communities and different depths.

6. Multiple contradictions were found in the following lines. Could the authors please fix these statements regarding the presence/absence of methanotrophs and methanogens so they are consistent.

Line 497-8 "Molecular analysis showed that both methanogens and methanotrophs were present at all four assessed sites Fig 6"

Line 666 "No aerobic or anaerobic methanotrophic prokaryotes were found in these peat deposits"

Line 610-611 "We conclude that in the observed absence of methanogenic and methanotrophic microbial populations, the in situ $CH_4$ observed in this study are trapped pockets of millennia old $CH_4$."

Line 672-4 "The absence of methanotroph activity is congruent with their absence in the results of 16S rRNA gene
amplicon sequencing and confirms that methanotrophic species are most likely not present or active in this
environment."

Line 780-782 "Whilst the source of $CH_4$ remains unconfirmed, we conclude that in the observed absence of methanogenic and methanotrophic microbial populations, the in situ $CH_4$ observed in this study are trapped pockets of millennia old $CH_4$."

Other contradictions:
Line 600-601 "Further, microbial analyses show that neither aerobic or anaerobic methanotrophic prokaryotes were activated by oxic or anoxic incubations. Therefore, we did not observe processes where biogenic $CH_4$ may have been produced in the present day."

7. Methanosarcinales are mentioned in the discussion but not in the results.

BTW: There are several recent studies showing the release of methane by bacteria and others.
-https://www.nature.com/articles/ngeo2837
-https://www.nature.com/articles/s41564-017-0091-5
-https://advances.sciencemag.org/content/6/3/eaax5343?
utm_source=TrendMD&utm_medium=cpc&utm_campaign=TrendMD_1

---

## Author Response (AR2)

The authors have done a lot of excellent work both responding to reviewers comments that will assist readers to understand their work. It is nicely rounded and very well placed into context.

**We thank the reviewer for the constructive feedback on our manuscript. Please find below an answer to the points for consideration.**

I have a few points for consideration on the manuscript.
1. The authors have an incorrect reference for the Bacterial reverse primer. Please see the following quotes from the Klindworth paper cited for the reverse primer:
"Per sample, two separate PCR reactions were performed in order to test two bacterial primer pairs for 16S rDNA amplification. Primer pairs were:
(i): S-D-Bact-0341-b-S-17, 5'-CCTACGGGNGGCWGCAG-3' (32), and
S-D-Bact-0785-a-A-21, 5'-GACTACHVGGGTATCTAATCC-3 (32);"
"32
Herlemann D.P.R., Labrenz M., Juergens K., Bertilsson S., Wani
ek J.J., Anderrson A.F.. Transition in bacterial communities along
the 2000 km salinity gradient of the Baltic Sea, ISME
J, 2011, vol. 5 (pg. 1571-1579)"

**We thank the reviewer for carefully checking this. The first publication of this primer is in Klindworth et al., 2013. We have not altered our in-text reference.**

**We agree with you that Klindworth et al., 2013 references this primer using Herlemann et al. 2011. However, we found no mention of S-D-Bact-0785-a-A-21, 5'-GACTACHVGGGTATCTAATCC-3 within Klindworth et al., 2013.**

2. In previous work by the authors in which these primers were tested they used a temperature gradient PCR to select 60° C. It is not stated what the authors observed at 60°C that helped them choose this temperature. Could the authors please include this information in the methods as it is relevant to the interpretation of the results and might help others in their lab work.

**We have included at L250: "The annealing temperature of 60°C was selected because we observed intense bands of the correct amplicon and no observable primer dimers at this temperature."**

3. Generally an annealing temperature of +/- 5° from the mean Tm of the primer pair, and as low as possible to achieve amplification is chosen. Typically this particular primer pair is run with annealing temperatures between 48°C and 56°C. The use of a higher annealing temperature could increase amplification bias reducing the phylogenetic range of template DNA amplified. Please carefully phrase any statements about the absence of detection of any clade as the absence could, more than usual, be due to amplification bias.
-https://academic.oup.com/femsec/article/60/2/341/584515
-https://academic.oup.com/nar/article/41/1/e1/1164457#119410076
-http://cshprotocols.cshlp.org/content/2009/4/pdb.ip66.short?
casa_token=h2nXrvzrHDIAAAAA:4IzeOXuwzgz48-
voKEUGXXMXq7jPlHPUUAkmR8WSgcMNN3B79NXvMPf_nkNgxkTC7SbgD5TJQ5I

**We agree that primer biases are inevitable with amplicon based studies, also due to primer mismatches or not binding primers due to suboptimal PCR conditions for certain microbial clades.**

**We have here chosen the PCR conditions based on the observation on the specific DNA samples. To further highlight this we have added the sentences to the section on amplicon sequencing at line 271: "We want to stress that biases associated with PCR-based amplicon studies can result in an observed absence of specific microbial clades and results need to be interpreted with care."**

4. Please provide standard details of the PCR e.g. volumes, concentrations etc of reactants and template, or provide a reference where these details are given.

**We have added these details to the section 2.6.2 Amplicon sequencing and analysis (Line 242): "The total PCR reaction volume was 25 $\mu$L with 1 $\mu$L forward and 1 $\mu$L reverse primer (20 $\mu$M), 1 $\mu$L template, 12,5 $\mu$L 2x concentrated SYBR™ Green PCR Master Mix (ThermoFisher, Carlsbad, CA, USA) and 9,5 $\mu$L sterile MilliQ."**

5. I have trouble reconciling the NMDSs in fig S7 which show a random placement of OTUs and figure 6 which shows that there is not just a difference in archaea and bacterial assemblages between sites but that differences can be seen at family and higher taxonomic levels and so should be very obvious at OTU level. Please provide details of how the NMDS were plotted as this difference seems strange. The statement that no difference in structure was observed according to the NMDSs while an accurate description of what is plotted (though because sites are not plotted on the NMDSs the statement is only partially supported), does not fit with the other analyses which show clear differences at sites in communities and different depths.

**We agree that the OTU figures do not provide a clear difference. This is mainly attributed to the fact that an OTU analysis does not necessarily visualize a difference in e.g. microbial family, which can be interpreted in taxonomic analysis. In addition, as stated in the materials and methods as well as in the legend description the OTU representation does not include rare taxa and is therefore only a coarse first overview of the population. We have therefore also decided to only supply these very rough overviews of the data in the supplementary data and to not draw any major conclusions on the data.**

6. Multiple contradictions were found in the following lines. Could the authors please fix these statements regarding the presence/absence of methanotrophs and methanogens so they are consistent.
Line 497-8 "Molecular analysis showed that both methanogens and methanotrophs were present at all four assessed sites Fig 6"
Line 666 "No aerobic or anaerobic methanotrophic prokaryotes were found in these peat deposits"
Line 610-611 "We conclude that in the observed absence of methanogenic and methanotrophic microbial populations, the in situ CH4 observed in this study are trapped pockets of millennia old CH4."
Line 672-4 "The absence of methanotroph activity is congruent with their absence in the results of 16S rRNA gene
amplicon sequencing and confirms that methanotrophic species are most likely not present or active in this
environment."
Line 780-782 "Whilst the source of CH4 remains unconfirmed, we conclude that in the observed absence of methanogenic and methanotrophic microbial populations, the in situ CH4 observed in this
study are trapped pockets of millennia old CH4."
Other contradictions:
Line 600-601 "Further, microbial analyses show that neither aerobic or anaerobic methanotrophic

prokaryotes were activated by oxic or anoxic incubations. Therefore, we did not observe processes where biogenic CH4 may have been produced in the present day."

**Thank you for carefully reading these lines. Indeed, no methanotrophs were found in the data (not in the activity data and no molecular signatures), and this was wrongfully stated in some sentences. It is our intention to highlight the observed absence of active methanogenic and methanotrophic populations. This we have stressed in the lines below. We have changed this in the relevant sentences as follows (see words highlighted in bold for additions and strikethroughs for deletions):**
Line 497-8 "Molecular analysis showed that  **methanogens** were present at all four assessed sites Fig 6"
Line 600-601 "Further, microbial **activity-based** analyses show that neither aerobic or anaerobic methanotrophic prokaryotes were activated by oxic or anoxic incubations. Therefore, we did not observe processes indicating biogenic CH4 may have been produced in the present day."
Line 620-621 "We conclude that in the observed absence of **active** methanogenic and methanotrophic
microbial populations, the in situ CH4 observed in this study are trapped pockets of millennia old CH4."
Line 675 "No aerobic or anaerobic methanotrophic prokaryotes were found in these peat deposits"
Line 672-4 "The absence of methanotroph activity is congruent with their absence in the results of 16S rRNA gene amplicon sequencing and confirms that methanotrophic species are most likely not present or active in this environment."
Line 780-782 "Whilst the source of CH4 remains unconfirmed, we conclude that in the observed **indicated** absence of **active** methanogenic and methanotrophic microbial populations, the in situ CH4 observed in this
study are **most likely** trapped pockets of millennia old CH4."

7. Methanosarcinales are mentioned in the discussion but not in the results.
BTW: There are several recent studies showing the release of methane by bacteria
and others.
-https://www.nature.com/articles/ngeo2837
-https://www.nature.com/articles/s41564-017-0091-5
-https://advances.sciencemag.org/content/6/3/eaax5343?
utm_source=TrendMD&utm_medium=cpc&utm_campaign=TrendMD_1

**Indeed we did not observe Methanosarcinales-sequences in the data. We have changed the sentence to indicate that Methanosarcinales are the most probably subject, but not observed in our data: "In contrast to $H_2/CO_2$ and acetate, methylated compounds are a non-competitive methanogenic substrate that is metabolized by *Methanosarcinales*. However, we did surprisingly not observe presence of this methanogenic family in our data."**

**We are indeed aware of the studies focusing on methane release by bacteria. However, on the bigger picture in these ecosystems it is unlikely that these micro-organisms do play a role (ecosystem fully anoxic, absence of labile organic matter). We therefore did not further speculate on potential roles of these micro-organisms in this ecosystem specifically.**

---

## Author Response (AR3)

**Response to reviewer comments**

We thank the Associate Editor for considering and reviewing the manuscript. We respond to the issue raised below.

**Comments to the Author:**
Thanks for submitting the point by point response to the reviewer's comments and revised text. I would request you to address the following issue.
It is impotant that the genetic sequence dataset, BioProject PRJNA639452, is made available since the Genbank provides the option of generating a link for reviewers and editors to check data deposits without making the data available to the general public. This is also important as far as BG data deposit policy is concerned. However, if you still unable to provide the data link, kindly furnish a clarification regarding the same.

*Author's Response:*

*We have now made the data publicly available and provide the following additional information:*

*Amplicon sequencing data were deposited in the GenBank database under the BioProject PRJNA639452 and can be accessed with SRA identifiers SRR12014238 - SRR12014261.*

*All amplicon sequencing datasets can be accessed via these SRA identifiers.*